# Biomedical Applications of Translational Optical Imaging: From Molecules to Humans

**DOI:** 10.3390/molecules26216651

**Published:** 2021-11-02

**Authors:** Daniel L. Farkas

**Affiliations:** 1PhotoNanoscopy and Acceleritas Corporations, 13412 Ventura Boulevard, Sherman Oaks, CA 91423, USA; dlfarkas@gmail.com; Tel.: +1-310-600-7102; 2Clinical Photonics Corporation, 8591 Skyline Drive, Los Angeles, CA 90046, USA; 3Department of Biomedical Engineering, University of Southern California, Los Angeles, CA 90089, USA

**Keywords:** optical bioimaging, spectral, multimode, superresolution microscopy, in vivo, coherence, cancer, neuroscience, Alzheimer’s disease, endoscopy

## Abstract

Light is a powerful investigational tool in biomedicine, at all levels of structural organization. Its multitude of features (intensity, wavelength, polarization, interference, coherence, timing, non-linear absorption, and even interactions with itself) able to create contrast, and thus images that detail the makeup and functioning of the living state can and should be combined for maximum effect, especially if one seeks simultaneously high spatiotemporal resolution and discrimination ability within a living organism. The resulting high relevance should be directed towards a better understanding, detection of abnormalities, and ultimately cogent, precise, and effective intervention. The new optical methods and their combinations needed to address modern surgery in the operating room of the future, and major diseases such as cancer and neurodegeneration are reviewed here, with emphasis on our own work and highlighting selected applications focusing on quantitation, early detection, treatment assessment, and clinical relevance, and more generally matching the quality of the optical detection approach to the complexity of the disease. This should provide guidance for future advanced theranostics, emphasizing a tighter coupling—spatially and temporally—between detection, diagnosis, and treatment, in the hope that technologic sophistication such as that of a Mars rover can be translationally deployed in the clinic, for saving and improving lives.

## 1. Introduction


*“It would be madness and inconsistency to suppose that things not yet done can be done except by means not yet tried.”*
Francis Bacon

Biomedical optical imaging [1] is a major area of research, with powerful new methods and compelling applications being introduced on an ongoing basis. Elegant as they may be, we believe that the ultimate way to judge the value of such laboratory advances is by the progress they can bring to fighting disease. In order to see the bench-to-bedside dream of translational research become a reality that would enable this, we need biophotonic approaches that, while technologically sophisticated, allow deployment into a clinical setting [2], due to their combination of safety, small size, relatively low cost, and high performance [3]. Our focus area is where light (an exceptional investigative tool) and patient meet [4], and improvements that yield better outcomes, by identifying and addressing obstacles preventing the timely clinical adoption of laboratory-based advances, not the least of which is the difficulty of detecting and characterizing very small entities (molecules, cells) within the human body [5], especially quantitatively, dynamically, and preferably without contrast agents. How and where we look becomes critically important, especially if one targets (as one should) early detection and theranostics; for this, new tools and strategies are needed, with likely new outcomes.

In this context, we review some of the advances that underscore the simultaneous need for new technologies, interdisciplinary understanding, and clinical relevance, focusing heavily on our own work, as we can follow the internal logic of its evolution, making it easier to draw conclusions that stress conceptual and methodological points, rather than dwell on details that are well fleshed-out in the literature cited. This should allow us to not only survey and evaluate the past but also set our sights on what could and should come next.

Bioimaging has evolved, within the past decades, from an ancillary research tool to a cutting-edge experimental method. Progress is driven by technological advances addressing well-articulated needs, and by developers and savvy investigators working together to translate deeper understanding into better treatments. Optical imaging, by far the oldest of bioimaging methods, makes us understand better by seeing better. Compared to other bioimaging, it is special in several ways: (1) it is intuitive and non-invasive; (2) it covers many orders of magnitude in time (~15), intensity (~12), and space (~8), while (3) having a very high resolution in these very same domains. Most importantly, given the diversity of biology, it is ideally suited to look, quantitatively, at the basic unit of life (the cell), in realistic environments (either in vivo or in laboratory conditions approximating this). Based on the many useful properties of light (intensity, polarization, coherence, wavelength, timing, interference, non-linear absorption), imaging contrast can be generated in a number of different ways that can be viewed as complementary, and even synergetic in the information they yield. Ideally, one would want to have as many of these options available, simultaneously and integrated, for investigating a biological problem of interest; while this is not easy (or easily affordable), one should work towards this goal. Even moderate progress in this direction should yield significant consequences—scientific, clinical, and commercial.

Optical imaging (performed with the powerful human eye-brain combination) was the first method of scientific investigation. In modern research, optical imaging is unique in its ability to span the realms of biology from microscopic to macroscopic, providing both structural and functional details and insights, with uses ranging from fundamental understanding to clinical applications.

Figure 1 summarizes our admittedly imaging-centered view of translational biomedical research. It is aimed to emphasize the disconnect between the various size and application domains. We believe that great benefit could be derived by designing and implementing tools that transcend a single type of application or organization level, especially if these new tools can be kept versatile enough to still address challenges in a specific, optimized way.

Advances in medicine and surgery depend strongly on our understanding of the human body’s anatomy and physiology, and of its biological/molecular underpinnings. What we do with this knowledge and how useful we can make it in healing patients depends not only on the physicians’ and surgeons’ skills but also on the technologies available to them. There is no discipline that brings these two critical components of healthcare together better and with more promise than biomedical imaging. The field has experienced explosive growth recently, as anatomical imaging is supplemented with dynamic, high specificity methods providing access to molecular mechanisms of relevant processes (senescence, apoptosis, immune response, angiogenesis, metastasis). The most advanced imaging is no longer only topological in nature, as most currently established imaging methods, but also provides molecular specificity, usually by labeling with appropriate biomarkers. Even more excitingly, optical molecular imaging has the potential of additionally delivering mesoscopic capabilities (i.e., microscopic resolution in macroscopic bodies in vivo), thus performing a role previously reserved for the gold standard in surgical decision-making, pathology.

We proposed and implemented a multimode approach to biomedical optical imaging at all levels, featuring hyperspectral imaging, and optimized for earlier, more quantitative and reproducible detection of abnormalities, and tighter spatio-temporal coupling between such diagnosis and intervention. Addressing major areas of unmet need in the clinical realm with these new approaches could yield important improvements in disease management. While the emphasis is on concepts and technologies, application-wise our work on cancer, stem cells, and neural processes (highlighting very early detection of Alzheimer’s Disease) will be reviewed, with emphasis on the new strategies needed to achieve the desired imaging performance, and their physics and engineering underpinnings. Thoughts about better ways for academia, the clinical, and the corporate world to work together for innovative biophotonic solutions and their use in addressing major diseases [4] will also be outlined.

## 2. Optical Microscopy and Its Applications


*“To develop drugs for people, we basically dismantle the system. In the lab, we look at things the size of a cell or two. We dismantle life into very small models.”*
Aaron Ciechanover (Nobel laureate)

Microscopy is an icon of the sciences, due to its history, versatility, and universality. Modern optical techniques brought to it such as confocal and multiphoton imaging provide subcellular-level resolution in biological systems. The integration of this capability with exogenous chromophores can selectively enhance contrast for molecular targets as well as provide functional information on dynamic processes such as nerve transduction. Novel methods integrate microscopy with other state-of-the-art technologies; nanoscopy, hyperspectral imaging, non-linear excitation microscopy, and optical coherence tomography all provide functional, dynamic, molecular-scale, and 3-dimensional visualization of important features and events in biological systems. Moving to the macroscopic scale, spectroscopic assessment and imaging methods based on properties of light and its interaction with matter, such as fluorescence, reflectance, scattering, polarization, and coherence can deliver diagnostics of tissue pathology, including neoplastic changes. Techniques that utilize longer wavelength photons allow exploration of processes that occur deep inside biological tissues and organs. A microscope, as a quintessential scientific tool, could cover every domain from single molecules (and their interactions) to living animals and humans, but in order to be the cutting-edge, multi-purpose device that can connect these domains, it needs to be very advanced technologically. Importantly, it has to *integrate* the imaging modes chosen into an instrument that focuses them onto the same specimen, in real-time. Our new concept for such an instrument relies on the multimode microscopy we introduced in the early nineties (see below), but takes it in the direction of affordability, user-friendliness, and possibly ubiquity by retrofitting, while simultaneously addressing all imaging challenges.

### 2.1. Confocal Microscopy

Confocal microscopy was invented by Marvin Minsky [6] to peer through the haze of neural tissue under a microscope. It improves the quality and axial resolution of microscopy by imposing spatial constraints on the photons reaching a detector, discriminating against contributions away from the plane of focus. Although the axial resolution improvement is only roughly two-fold, this was a major advance in 3D microscopic imaging, becoming commercially available—not surprisingly—in the eighties, after the original patent expired. Excellent studies and visualizations of biological tissue were and continue to be achieved using this method [7], and quantitation is also improved since the voxels probed are reduced in size (especially depth). Since it is mostly implemented as a point scanning method, with an intense laser source as excitation for the specimen’s fluorescence, the image acquisition speed is not very high, and photobleaching could constitute a problem [8]. With new fluorescent probes, better detection schemes and other improvements, such as spectral options, these issues have become less of a hindrance in carrying out outstanding research, including in live cells and tissues. We will highlight here two new methods we developed, enabled and enhanced by the confocal capabilities.

#### 2.1.1. Membrane Electrical Potentials

We became interested in a new application for digital imaging microscopy, and particularly its axially improved confocal version, wondering whether one could quantitate an important physiological feature—membrane electrical potential—in living cells, where biochemical manipulations allow intentional, well-understood changes to be induced to probe mechanisms.

At a time that biologically-relevant cell bioelectric features were starting to be measured by fluorescence, especially in a dual-wavelength ratiometric mode [9], we started from a different observation (Figure 2):

For living cells, the distribution of charged, membrane-permeable molecular probes such as hydrophobic cationic dyes like tetramethylrhodamine ethyl ester (TMRE) or methyl ester (TMRM) [12], between an external aqueous medium, the cytoplasm, and the aqueous compartments of intracellular organelles like the mitochondria is governed by the relative membrane electrical potentials of these regions through coupled equilibria described by the Nernst equation. The charged, low membrane binding and toxicity rhodamine dyes [12] allowed monitoring of these equilibria through digital imaging microscopy. We employed this combination of technologies to assess, simultaneously, the membrane potentials of cells and of their organelles in situ, by following their fluorescence, with a good temporal resolution, in several cultured cell lines [11]. At equilibrium, these Nernstian dyes would accumulate about 10-fold in the cytoplasm and about 5000-fold within mitochondria in a cell with a membrane potential of −60 mV and mitochondrial membrane potential of −150 mV (see Figure 2). The time course of variations induced by chemical agents (ionophores/protonophores, uncouplers, electron transport, and energy transfer inhibitors) in either or both these potentials was easily quantitated, and in accordance with mechanistic expectations [11]. The reason that this fluorescence spectroscopy method works in cells is that the focused beam of a microscope creates within the cells a virtual microcuvette that is, in its entirety, contained within the cell body. The same is not true of the much smaller mitochondria: their sub-micron dimensions do not allow complete containment of this virtual microcuvette, and thus the fluorescence intensity is diluted by contributions from regions outside the organelle. Therefore, in order to get not merely a lower-limit estimate of the mitochondrial membrane potentials measured, one needs to address this issue. We used the better axial resolution and thus optical sectioning ability of confocal microscopy to re-measure them, and found higher values, around −170 mV. To our knowledge, this constituted the first use of confocal microscopy for non-invasive quantitation of an important electrophysiological functional parameter in living cells. Subsequent work [13] allowed a proper calibration/correction of these membrane potential values, taking into account the 200–300 nm dimensions of typical mitochondria. In the intervening years, TMRE and TMRM have become the mainstay of these types of measurements, and the methodology remained applicable to the study of more subtle and specific, biologically induced functional membrane potential changes in cells. More recently [14], superresolution microscopy (see below) has been used to visualize the inner mitochondrial membrane and showed that cristae membranes possess distinct mitochondrial membrane potentials, representing unique bioenergetic subdomains within the same organelle, making it possible to calculate membrane potentials of individual cristae, found to exceed −180 mV. With all this, more interesting and biomedically relevant questions can be addressed, such as the importance of membrane potentials and their heterogeneity in cancer [15] and their role in neurodegeneration [16].

#### 2.1.2. Topologically Resolved Epigenetics by 3D Quantitative DNA Methylation Imaging

We used a high-end commercial (spectral) confocal microscopy workstation (Leica Microsystems TCS SP5X Supercontinuum, Mannheim, Germany) to investigate the feasibility of using a high-resolution optical imaging and new analysis software we built to study epigenetics at the subcellular, even subnuclear level, specifically DNA methylation. We aimed to assess the correlation of in situ global DNA methylation features, such as nuclear load and spatial distribution of methylated cytosine (MeC), to the phenotypic elements that distinguish one type of cells from another. All cells of a given organism share the same genotype, but not all cells have the same phenotype. These differences arise from cell-specific gene expression programs that are modulated to a large extent by epigenetic signatures, such as DNA methylation and histone modification. These modifications influence chromatin conformation and are involved in regulating what part of the genetic code is transcribed to messenger RNA (mRNA) that could be later translated into proteins. So far, DNA methylation is known to be the most stable epigenetic modification that affects the phenotype of a cell, due to its role in the concerted promotion and/or suppression of imprinted genes, the long-term silencing of repetitive DNA elements, and the inactivation of a single copy of X-chromosomes in females. Aberrations in DNA methylation patterns have been identified in cancer cells and other cells exhibiting complex human syndromes, and, therefore, distinct differences exist between methylation patterns in “healthy” cells compared to “diseased” cells [17,18,19,20,21]. A better understanding of these differences through studies of DNA methylation profiles may lead to breakthroughs in disease diagnostics and prognostics.

Historically, the vast majority of studies on DNA methylation have focused on the molecular profiling of large volumes of cells, trying to determine the locations of MeCs within sequences of certain gene promoters and other specific DNA fragments of interest (see discussion in [21]). This type of analysis has been very important in understanding the mechanisms of methylation and their effects on cellular functions. Researchers around the world are now trying to characterize the methylation patterns of entire genomes using holistic molecular methods, much like the sequencing of the human genome through the Human Genome Project, and publications have been made showing differential MeC profiles (methylomes) in the context of cellular differentiation and various malignancies [22]. Genome-wide, molecular-based analyses using DNA microarray and next-generation sequencing technologies are of great benefit in understanding the progression of certain diseases and developing therapeutic strategies. However, they are currently challenged, in terms of specificity and sensitivity, when it comes to the characterization of limited amounts of DNA from single cells and are so far applied to obtain only averaged methylation profiles across batches of cells. Since most complex diseases, such as different cancer types, are comprised of a heterogeneous mixture of cells with diverse methylomes, a cell-by-cell analysis of diseased tissues has the potential to improve the quality of information gained through DNA methylation profiling.

Furthermore, current molecular-based analyses are time-consuming and costly, when utilized in cell-by-cell DNA methylation profiling. However, this level of scrutiny is necessary to avoid the loss of cell-specific information that mostly occurs when large pools of cells are analyzed as a batch. Therefore, there is a strong need for a rapid and cost-effective assay that performs global DNA methylation analysis at the single-cell level, towards the goal of improving diagnosis and prognosis in oncology. It is with this motivation that we developed an imaging-based, quantitative, and cytometric analysis of DNA methylation, and we named it *3-Dimensional Quantitative DNA Methylation Imaging* (3D-qDMI) [17,18,19,20,21].

Recent advancements in cellular imaging and computation have allowed for large volumes of images to be analyzed in a reasonable amount of time at substantially lower costs. In contrast to molecular-based analyses of DNA methylation patterns, imaging-based analysis techniques would allow a rapid cell-by-cell characterization of diverse cell populations that have been grown under different conditions. We believe our method could improve cancer diagnostics, prognostics, and post-operative patient monitoring when translated into the clinical setting. Moreover, imaging-based methylation analysis could be used in pre-clinical studies of epigenetic drugs to better illuminate the effects these drugs have on the epigenetic signatures of cells, which are connected to the higher-order genome organization in the nucleus [22]. 3D-qDMI could also be performed in combination with downstream molecular analysis of selected cells to correlate sequence-based MeC profiles and nuclear MeC topology to investigate the relationship between “epi-phenotypes” and “epi-genotypes” at the chromatin level.

Briefly, the methods for 3D-qDMI and subsequent imaging-based analyses were as follows: Immunofluorescence-labeled specimens were generated from cells and/or tissues of interest, using a specific antibody targeted to nuclear 5-methylcytosine. To delineate the inhomogeneous distribution of overall DNA in the nucleus, specimens were counterstained with 4′,6-diamidino-2-phenylindole (DAPI), a non-specific dye that intercalates in the DNA double helix. The specimens were then imaged at high resolution using laser-scanning confocal microscopy. The DNA methylation patterns in each collected 3-D image were then analyzed with an image analysis software package that we specifically developed for this microimaging-based approach.

Three different analytical modules have been utilized to correlate DNA methylation features, such as the overall load and the spatial co-distribution of MeC and genomic DNA (represented by DAPI) in single human cells, with cellular proliferative behavior. The first module assessed feature similarities of each cell to the overall population using statistical comparison of the co-distribution of DAPI and MeC signals. Using these similarity measures, the overall homogeneity of cell populations could be determined. One major advantage of the imaging-based approach—previously not available in molecular-based approaches—is the identification of outlier cells or cells that can be either eliminated from further analysis or analyzed separately to improve confidence in the analytical outcome. The second and third modules focused on identifying topological variations in global DNA methylation levels and genomic organization, especially within the euchromatic and heterochromatic regions of nuclei. This novel approach to cellular characterization is not possible with currently available molecular methods. Therefore, an imaging-based cytometrical analysis may add valuable information regarding chromatin structure and positioning of methylated sites that are relevant to disease diagnostics and prognostics.

Our publications using parts of these analytical modules have clearly demonstrated that differences in nuclear methylation patterns can be identified between cancer cells treated with epigenetic drugs and those cultured in parallel as untreated controls [17,19,21]. Since treatment with epigenetic drugs not only affects the DNA methylation of cells but can also influence cellular growth behavior [23], it is conceivable that differential MeC patterns could exist in cells of different phenotypes.

To validate the hypothesis that 3D-qDMI can be used to distinguish and discriminate cells with different proliferative capacities, based on their nuclear DNA methylation patterns, the following feasibility studies were performed [17,18,19,20,21,24,25,26]: (a) Correlation of cellular growth with DNA methylation patterns; (b) comparison of DNA methylation patterns in normal, primary cells as they progress to replicative senescence and hyper-proliferating immortal cancer cells; (c) comparative analysis of DNA methylation patterns between primary and cancer cells after accelerated growth arrest in vitro; and (d) assessment of DNA methylation patterns in cells at different cell cycle stages.

The results gathered from these studies can be compiled into a database that can be used as an assessment tool of cellular phenotypes in translational medicine, with the ultimate goal of creating a novel quantitative method that will be used by pathologists in oncologic diagnostics and in the discovery and development of anti-cancer drugs that target epigenetic modifications in the genome.

3D-qDMI was not designed to replace molecular-based assays, as it cannot identify precise levels of gene promoter methylation nor quantify the absolute number of MeC residues in nuclear DNA. Nevertheless, it can provide information regarding the relative in situ concentration and spatial distribution of MeCs. Moreover, for the purposes of pathology, a more pertinent question may be whether “healthy” cells can be discriminated and distinguished from “diseased” cells and phenotypic composition of cell populations (such as in tissues) can be determined, to which this imaging-based assay is focused. Thus, 3D-qDMI intends to address the need for a DNA methylation assay that can be utilized in translational medicine and, ultimately, transferred to the clinical setting. It is indeed a completely different approach to assessing variations in DNA methylation profiles and will bring a new perspective to the understanding of how epigenetic differences affect cellular phenotypes in the field of pathology.

Much of the cellular phenotyping methods for tumor cells in biopsy samples currently rely on a static, morphology-based evaluation by a trained physician, which in many cases do not sufficiently explain the heterogeneous cellular composition of tumors or their differing growth potentials. The novelty of this work consisted mostly in the way the data were analyzed (summarized graphically in Figure 3), geared towards delivering a more dynamic and activity-based picture of cells and may therefore allow more predictive evaluation of cell population behavior, based upon quantifiable measures of function-related chromatin texture. Moreover, these studies are consistent with the general tenets of cancer treatment: the identification of cells with abnormal growth behavior and their reprogramming and/or extinction through various types of therapeutic treatments, including surgical removal.

As epigenetic mechanisms, such as DNA methylation, play important roles in oncogenesis and cancer progression, the advent of cellular phenotyping via quantitative DNA methylation imaging is a promising approach to improve cancer diagnostics, prognostics, and post-operative patient monitoring. The implications for drug screening are also intriguing and were reviewed in [20].

Taken together and followed up by subsequent studies on stem cells and prostate and lung cancer [24,25,26], the studies reviewed show the feasibility of using a high-resolution confocal imaging-based approach to assess DNA methylation profiles for important clinical applications.

### 2.2. Standing Wave Microscopy: Towards Omnidirectional Superresolution

The resolution of a light microscope is limited by diffraction. Traditionally, the in-plane (X,Y) resolution limit is defined by the Rayleigh criterion (as also captured in Abbe’s similar formula):*R* = 1.22λ/(2*n* sin α) = 0.61λ/*NA*
where λ is the wavelength, *n* is the index of refraction, and α is the half-angle of the maximum cone of light that can enter the entrance aperture of the objective lens. The term *n* sinα is the numerical aperture, NA. For example, in the context of light microscopy, a 63 × oil immersion objective with a NA of 1.4 operating at a wavelength of 450 nm may resolve two points if they are at least 196 nm apart. The resolution in the third or axial dimension is approximately four times worse. Many of the features of interest in fluorescence microscopy of cells are not resolved by a conventional optical microscope. This represents a fundamental barrier to progress, for example in cancer research where imaging is used to study changes in cytoskeletal, membrane, and chromosome structure, and to visualize changes in DNA, such as patterns of methylation (see Section 2.1.2 above). The recent proliferation and recognition of super-resolution methods (Method-of-the-Year, Nature Methods, 2008; well-deserved Nobel prizes to Drs. Stefan W. Hell, Eric Betzig and William E. Moerner [www.nobelprize.org/prizes/chemistry/2014/summary/, accessed 11 April 2021]) reflects the realization of this need.

Super-resolution microscopy methods (also known as nanoscopy and as point-spread-function engineering) can be classified into several approaches.

(1)In the first class are methods (STED, SAX, SPEM, RESOLFT, PALM, and others) that manipulate the signal generation within the sample. For example, in photoactivated localization microscopy (PALM), the density of fluorophores is controlled so that one light emission point is activated at a time within the volume covered by the point spread function (PSF) of the imaging system. Under these conditions, the location of the fluorophore can be determined to a very high precision, which is only limited by the signal-to-noise ratio of the camera. Repeatedly activating different sets of fluorophores allows the assembly of a high-resolution image from the individual point source location maps. Other methods in this class use the saturation of fluorophores, non-linear quantum-effects such as the stimulated emission depletion (STED) via a second illumination wavelength, or the blinking of emitters (quantum dots). *All* these methods require extreme stability of the microscope, long acquisition times, specialized protocols and/or fluorophores, and/or high-power illumination sources that can cause severe photobleaching. These have been reviewed extensively [27,28,29,30,31], and have been constantly improved and supplemented, yielding impressive performance (see e.g., [32]).(2)In the second class, methods try to subdivide the PSF by using specialized illumination systems that use interference effects to narrow the point spread function of the microscope. As will become apparent below, structured illumination systems effectively increase the numerical aperture to improve resolution. One approach to increase the numerical aperture is to use two objectives to observe the sample from both sides simultaneously. If both images are combined optically and in a coherent fashion, the effective numerical aperture is doubled, which leads to 4π microscopy. Unfortunately, this is achieved with extreme alignment difficulties, which makes 4π microscopy rather impractical and expensive (see [27,30,31]).

An alternative is standing wave microscopy (SWM), which we pioneered [33,34,35,36], and believe to be the first PSF engineering super-resolution method published. In its simplest embodiment (see the leftmost image in Figure 4, a/top), a mirror is placed directly behind the specimen—illuminated through the microscope objective lens—in an epi-fluorescence microscope. The light passes through the sample under investigation and is reflected back towards the objective lens. Thus the illumination light is traversing the sample twice, once from the objective lens towards the mirror and once in the opposite direction, and an interference pattern that is periodic (standing wave) along the optical (Z) axis will be observed (with the added advantage that the pattern will always have a node at the mirror, thus allowing the precise axial location of the fringes, which have peaks spaced λ/2n cos θ apart, with θ the angle of the counterpropagating beams, in this case, 180°). The important property of this interference pattern is that its period is less than half of the wavelength of the excitation light; this is used to significantly increase the axial resolution of the microscope. We obtained axial resolutions of better than 35 nm (Figure 4, second panel from left, [33]), and subsequently as good as 10 nm [34,35,36].

The primary limitation of standing wave microscopy stems from the fact that the interference pattern is produced by two counter-propagating nearly planar wavefronts. Thus, the interference pattern is periodic along the *Z*-axis only and has no significant structure in the X and Y axes. Therefore, only the axial resolution is (significantly) improved, while that along the X and Y axes remains unchanged.

Our newly developed omnidirectional standing wave microscopy (OSWM) method, proposed by Dr. A.G. Nowatzyk [37], addresses these limitations and has not been presented before. Depicted in Figure 4 (right) is a standard oil-immersion objective, but newer objectives corrected for water or glycerol immersion work equally well. The sample is affixed to the top surface of the cover-slip and is immersed in water or a liquid with an index of refraction that matches the embedding medium. The fluorescent emissions from the sample are observed by the objective lens. The emitted light passes subsequently through a dichroic beam-splitter and the emission filter, which blocks reflected excitation light. The tube lens forms a virtual image, which is projected with a relay lens onto a CCD image-sensor that records the image. What has been described so far is the optical path of a standard, commercial, off-the-shelf microscope that is operated in the epi-illumination fluorescence mode.

For OSWM microscopy, a Reflective Diffractive Optical Element (RDOE) is added just behind the sample. The RDOE is mounted on a three-axis translation stage, which allows the RDOE to be moved in the X, Y, and Z axes. The position resolution of the translation stage must be small compared to the wavelength of the excitation light. For example, commercial piezoelectric actuators with capacitive feedback can achieved repeatable, controlled motion on a sub-nanometer scale, which is sufficient for OSW microscopy. Like the mirror in the previously developed standing wave microscope, the RDOE reflects the excitation light back towards the objective lens and creates an interference pattern with the incident excitation wavefront throughout the sample volume. However, unlike the interference pattern created by a plane mirror, the interference pattern created by the RDOE has a complex, three-dimensional structure with sharp contrast in all three dimensions. This interference pattern is a function of the position of the RDOE, which can be moved over the sample volume in a tightly controlled fashion. In Figure 4, the right panel is the intensity distribution in the XZ plane of an illumination beam that originates from a circular aperture at the bottom of the panel. Here, the typical near field diffraction pattern of a circular aperture appears. The middle panel shows the intensity distribution of the illumination beam when it is reflected by the RDOE, which is an array of pyramidal reflectors in this case. The top-right panel shows the interference pattern that is created by the two counter-propagating wavefronts. An axial intensity modulation is produced that has a period of approximately one-half of the excitation wavelength. The bottom set of panels shows the intensity distribution in the XY plane, at a position that is indicated by the white line in the top-right panel.

In image reconstruction, the computational requirements for OSWM are much larger than those for ordinary SWM. In SWM, the interference pattern has a very regular structure that leads to a relatively simple, direct mathematical formulation that can be solved directly. The required operation to combine the three images that SWM acquires for each focal position requires less than 100 arithmetic operations per pixel, which any contemporary PC can perform in about the time it needs to load the images. Essentially, each XY plane along the *Z*-axis is processed independently. In OSWM, there is no easy mathematical structure and the image planes along the *Z*-axis are coupled and cannot be computed without taking the results from the adjacent planes into account. This is the price to pay for relative optical simplicity. There are several established algorithms to solve the generalized inverse Radon transform that forms the core of OSWM image reconstruction. In this instance, an algebraic reconstruction based on preconditioned conjugate gradient method is the most practical technique to use for image reconstruction. The availability of high-performance computing based on graphics processing units (GPUs, such as those from Nvidia) allowed us to better address this challenge.

It is important to note that OSWM offers super-resolution in all three spatial directions and that it is possible (indeed recommended) to implement it as a relatively easy add-on to existing microscopy workstations, especially since most of the advanced systems already have lasers, advanced optics, and high-end computers. We plan to present our detailed implementation, including the computational part, and imaging results obtained with this technique elsewhere (A.G. Nowatzyk and D.L. Farkas, in preparation). Additionally, it is also worth stressing that this approach can be combined, both in principle and in practice, with other advanced microscopic imaging methods, such as STED or even multiphoton microscopy.

One specific application of OSWM we intend to focus on is the imaging of DNA methylation on a cell-by-cell basis. As discussed above (Section 2.1.2), epigenetic mechanisms, such as DNA methylation play a key role in cellular differentiation, and imbalances in methylation patterns are associated with a variety of complex diseases, including cancer. The approach of quantitative analysis of the differential distribution of DNA methylation in single-cell nuclei of large cell populations should provide comprehensive information that will benefit basic research of the functional 3D genome architecture and translational areas of cancer pathology and treatment [17,18,19,20,21,24,25,26]. Our ultra-high resolution OSWM optical system will enable the transformation of DNA methylation imaging from a currently qualitative method into an emerging quantitative technology, producing image-based methylation profiles as bio-signatures with diagnostic and prognostic value. Thus, the availability of OWSM could have a fundamental contribution to the quickly growing research field that combines the multidisciplinary efforts required for producing fast, automated image-based techniques that allow high-throughput, high-resolution epigenetic screening of mammalian cells with long-term potential benefits including:

(a) finding targets for epigenetic treatment/predicting cancer therapy responsiveness;

(b) assessing environmental factors that impact the epigenomic makeup of cells;

(c) characterization of complex epigenomics related diseases on a cellular basis;

(d) enabling the use of cell models in drug development (pharmacoepigenomics).

Another research area that could greatly benefit from omnidirectional super-resolution capabilities is virology, especially as it pertains to mechanisms, studied in live cells (see Section 6, Discussion).

### 2.3. Multimode Microscopy

The cell is the basic unit of life, and it is highly desirable to capture its makeup and activities in conditions that approximate its natural state, i.e., inside the body, functioning—normal or abnormal, as dictated by circumstances [38]. We started with a relatively simple concept and built a multimode microscopy workstation that allowed this [39] by focusing on live-cell imaging, thus targeting a narrow window of opportunity. The cells were “tricked” into feeling as though they were in the body (by using accessories such as the environmental control live-cell chamber [9] allowing strict temperature control and laminar flow of nutrients and other perfusates), and were observed during their natural functions, such as cell division, or reacting as a consequence of induced perturbations of interest (physical, chemical or biological). We aimed to gather as many different types of imaging data as possible, as efficiently as feasible, targeting their complementary properties/behavior (e.g., morphology, adherence, motion, molecular interactions and concentrations, ionic fluxes, etc.). The implementation was not so simple: in spite of a solid microscopy platform, detectors [40], light sources, opto-mechanical components were not easy to bring together and automate as add-ons, under user-friendly software control. The choice of the modes implemented concentrated, to a large extent, on traditional modes of microscopy, supplemented with newer methods that came into the field within the previous decade (ratio imaging, multicolor imaging, fluorescence recovery after photobleaching, reflection interference and TIRF, confocal microscopy, 3D computational deconvolution). Multimode imaging data were linked into multi-location time-lapses, for complete 4-D datasets (see Figure 5). We focused on topologically quantitative comparisons between the imaging data obtained by the various modes [41], improving data handling performance and automation [42,43], including telepresence uses [44], and on important applications in molecular, cellular, and developmental biology (see below, Section 2.3.1), as well as toxicologic pathology (on live cells) [45] and the brain and the nervous system, including development and abnormal states, assessed at the live-cell level [46].

Given the amount and diversity of the data gathered, we also focused on machine vision and robotics issues [45,46,47], new types of image handling, from better remapping [44] and segmentation to machine learning and AI approaches [47,48], better 3D methods [49] and color/wavelength multiplexing (see Section 2.5 below). All major microscope manufacturers worked with us, and eventually added significant automation to their top-of-the-line instruments, thus adopting multi-mode microscopy. Additionally, high content screening, a very useful new field based on a systems cell biology approach to drug discovery has its origins—technological and conceptual—in these developments [50].

Two applications will be discussed below, cytokinesis and cancer stem cells.

#### 2.3.1. Live Cell Motion

Since one of the most characteristic features of live cells is their movement, whether to perform basic functions such as cell division or respond to stimuli, from biological to chemical to mechanical (e.g., wound healing), we explored a lot of these using the new capabilities afforded by the multimode microscope. For example, the mechanisms of cytokinesis in cell division have been difficult to zero in on because of the short duration and complex spatio-temporal dynamics involved in the formation, activation, force production, and disappearance of the cleavage furrow. This was investigated [51] in a seminal article (paper-of-the-year in *Mol. Cell. Biol.*) taking full advantage of the multimode features, with the quantitative, topologically resolved intracellular dynamics of myosin II being elucidated and found to be very similar to those in wound healing, at least in fibroblasts. A representative set of images is shown in Figure 6, with the mechanistic conclusions in schematic form.

Another important application of multimode imaging (in the realm of cytokinesis) is the measurement of the traction forces involved in cell division. Cells dividing in culture undergo a dramatic sequence of morphological changes, redistribution of actin, myosins, and other molecules into the cleavage furrow, and re-spreading before daughter cells finally separate at the mid-body. Knowledge of forces governing these movements is critical to understanding their mechanisms, including whether the formation of the cleavage furrow results from increased force generation at the equator or relaxation at the poles. Researchers at our national center [52] have quantitatively mapped traction forces in dividing cells, by extending the compliant silicone-rubber substratum method to detect minute forces down to nanonewtons, using a new silicone polymer to fabricate substrata whose compliance could be adjusted precisely by ultraviolet irradiation and calibrated opto-mechanically. They showed that traction force appears locally at the furrow in the absence of relaxation at the poles during cleavage. Force also rises as connected daughter cells re-spread and attempt to separate, suggesting that tension contributes to the severing of the intercellular bridge when cytokinesis is completed. These types of studies would not have been possible without the multimode capabilities.

Subsequent uses of different flavors and implementations of multimode microscopy have a common theme: focusing on those complex, challenging questions that require the sophistication of multiple modes, deployed simultaneously and in concert, for providing the answers [53].

#### 2.3.2. Cancer Stem Cells

Glioblastoma multiforme (GBM) is typically comprised of morphologically diverse cells within the tumor mass. Despite current advances in therapy, the morbidity and mortality of GBM remain very high [54], due at least in part to the focus of most treatments on the bulk of the tumor. However, the diverse cells within GBM may play different roles in tumorogenesis. We set out to study this with the aid of multimode microscopy since there was increasing evidence that cancers might contain and arise from stem cells. GBMs contain cells that express neural markers as well as cells that express glial markers, indicating that there may be multipotent neural stem cell-like cells. Such mixed glioblastomas may develop from neural stem cells (NSCs) or from differentiated cell types that acquired multipotential stem cell-like properties by either reprogramming or de-differentiating in response to oncogenic mutation. We were able to demonstrate [55] that human adult GBMs contain a subpopulation of cells that can form neurospheres in a defined stem cell medium with growth factors. The spheres share many characteristics of stem cells, including self-renewal ability and multipotent differentiation, which can produce daughter cells of all phenotypes present in the GBM. Furthermore, we showed that these spheres are different from normal neurospheres, being able to reform new spheres after the induction of differentiation. More importantly, after in vivo implantation only the isolated tumor stem cells were able to form tumors that contained both neurons and glial cells. This suggested that a subpopulation of cells exist within adult GBM, which may represent a general source of cancer stem cells in adult brain tumors and need to be targeted for more effective and specific cancer therapy.

Cancer stem cells are potentially important because it is likely that the stem cell population within tumor mass plays a key role in the recurrences that occur after current treatments. Thus, it is critical for cancer therapy that treatments must target and eliminate this special population of cancer cells. Consequently, the need to identify and study cancer stem cells becomes significant. Our data indicated that human adult GBMs contain a subpopulation of cells that can self-renew and differentiate into mature cell types, recapitulating the diverse complexity of primary GBMs. Several lines of evidence supported that we have isolated cancer stem cells from adult human GBMs: (1) the isolated cells only account for a small fraction in tumor mass, form spheres that are morphologically indistinguishable from normal neurospheres, and express known NSC markers; (2) the cells can self-renew and proliferate to generate sub- spheres and different progenies; (3) the isolated single mother cell can differentiate into multi-lineage progenies; (4) spheres derived from a single mother cell possess a different phenotype compared with normal NSCs upon differentiation, can reform spheres after the induction of differentiation, and display genetic aberrations commonly found in brain tumor cells; (5) the isolated neurosphere-forming cells can produce brain tumors in nude mice, whereas the non-sphere-forming monolayer cells do not. These main findings are illustrated in Figure 7.

In addition to providing further evidence for the existence of cancer stem cells in solid tumors, as the presence of stem-like precursors in adult human glioblastomas has been independently reported after our work was completed (Galli et al., 2004), our finding that cancer stem cells can reform new spheres after the induction of differentiation has an important implication. It may, for example, provide a basic strategy for identifying cancer stem cells from normal NSC. We found that glioblastoma spheres derived from a single mother cell can differentiate into the three CNS cell lineages and recapitulate the phenotypically complex property of the parental tumor. These data suggest that the presence of varied cell types within the tumor is not simply a consequence of different types of cells growing together to form tumor mass, but rather is an intrinsic property of cancer stem cells. The isolated brain tumor stem cells, not only giving rise to further stem cells but also to a diverse population of other phenotypes, may serve as a useful object for understanding the fundamental similarities and differences between normal neurogenesis and tumorigenesis in CNS.

An interesting implication arises from the ability of one glioblastoma cell to generate neurons as well as astrocytes and oligodendrocytes. These cells were isolated from adults, suggesting that there are two possibilities. On the one hand, these might be converted from normal NSC by oncogenic stimulation. They were stem cells before acquiring the tumorigenic property. On the other hand, these cells might be somehow either reprogrammed or de-differentiated from a more differentiated cell type in response to some signal transduction cascades. Studying the molecular basis involved in the oncogenic conversion of stem cells or the conversion from more differentiated cells into a stem cell-like phenotype might shed light on the biology of both NSCs and cancer cells [56].

We believe this study highlighted the importance of cancer stem cells in brain tumor research and suggested a basic strategy to identify brain cancer stem cells from normal NSCs. The study also indicated that the cancer stem cells of adult brain tumors may serve as a tool for studying the basic biology of adult stem cells, with optical imaging—represented by multimode microscopy—as an important experimental tool.

Our further investigations [57] revealed that one can isolate tumor stem-like cells from benign tumors as well, an unexpected result that cautions against simplistic interpretations of these very complex phenomena. The field remains very active, understandably so in view of the great potential implications for the treatment of important cancers [58].

### 2.4. Hyperspectral Microscopy for Clinical Diagnostics

In exploring the various imaging modalities that we were to deploy in the study of cells and tissues, especially in situations where there were several entities to be monitored and quantitated, distinct from one another, we found that the most discriminating method is spectral imaging. When light impinges upon an object, it can be re-emitted, reflected/scattered, transmitted, or absorbed. Spectroscopy measures these phenomena in the temporal or frequency domains in order to determine important physical properties of the object being probed. Depending on the energy of the light and the nature of the object, some of these interactions, such as absorption by particular chemical bonds, can reflect the presence and quantity of certain well-understood constituents. The addition of imaging to spectroscopy can trace its origins to pioneering efforts in airborne satellite-based remote-sensing, which was developed on the premise that information present in the optical properties of natural and man-made objects could be used to detect and monitor these entities from a distance. Similar premises are at the basis of non-contact studies by biological spectral imaging, particularly with “optical biopsy” [59]. Optical imaging uses light to probe a scene, in order to determine structure and organization. Unlike spectroscopy, in its simplest form, it does not probe fundamental chemical or physical properties; rather, it presents data to our human visual system for perception and understanding. Thus, while there is a connection between image and content, it is not easy to combine the two rigorously, and often analysis may address spectral and spatial content sequentially rather than simultaneously. While spectroscopy and imaging have been coupled in the past, usually this has involved obtaining a point or a line of spectroscopic information out of an entire two-dimensional image. Spectral imaging has been brought to biology, in an effort to determine, with high spatial resolution, not only what a scene “looks like” but also what it contains, and where the various classes of objects are located. Human color vision is, of course, a form of imaging spectroscopy, by which we determine the intensity and proportion of wavelengths present in our environment. Spectral imaging improves on the eye or color cameras in that it can break up the light content of an image not just into red, green, and blue, but into an arbitrarily large number of wavelength classes. Furthermore, it can extend the range to include the invisible ultraviolet and infrared regions of the spectrum denied to the unaided eye; this type of imaging is usually known as hyperspectral. The result of (hyper)spectral imaging is a data set, known as a data cube, in which spectral information is present at every picture element (pixel) of a digitally acquired image, whether subcellular or planetary. Integration of spectral and spatial data in scene analysis remains a challenge but can yield rewarding results.

We deployed numerous versions of spectral imaging, based on well-understood physics (multi-filters, continuously variable filters, prisms, liquid crystal tunable filters, snapshot spectrotomography, Fourier transform Sagnac microinterferometry, acousto-optic tunable filters, and Fabry-Perot constructs) to all areas of biomedical imaging, and have reviewed our results and the field extensively [59,60,61,62,63,64]. Therefore, basic concepts and technologies will not be discussed here; rather, a number of interesting applications ranging from cyto- and histo-pathology to in vivo and intrasurgical imaging will be reviewed, in roughly ascending order of difficulty and needed innovation.

#### 2.4.1. Cytopathology

In cytopathology, one of the oldest and most broadly utilized tests is the Papanicolau (“Pap”) smear for detecting cervical cancer. It relies on a specimen that consists of a single layer of cells, stained with a well-trusted, colorful set of dyes, and the evaluation is usually by a trained cytopathologist, using a manually operated optical microscope. Unfortunately, due to this setup, there are many problems with the test. To name a few, (a) throughput is low, due to lack of automation; (b) the calls by the cytopathologist are subjective, and made even more difficult by sample issues (e.g., cell clumping) and fatigue (performance gets lower towards the end of the workday), (c) lack of a digital record (at least for comparison with a preceding or future test on the same patient), and so on. We addressed some of these issues by using an automated microscope with spectral imaging capabilities and found that the evaluation of the specimens (still, of course, in the presence and with the help of a pathologist), as aided by spectral-assisted segmentation, is much improved. In Figure 8, one can see that even simple analysis differentiates normal cells from dysplastic, intermediate, and superficial squamous ones, as well as identifies others that are not necessarily expected to appear in the field of view (lymphocytes, polymorphonuclear cells). Notably, a known added advantage of hyperspectral imaging (the fact that, having come from satellite reconnaissance, the method was thought about and custom software-analyzed by very capable and well-funded people) was borne out by our experiment: an off-the-shelf spectral analysis package (ENVI, L3Harris, Broomfield, CO, USA) used in geophysics/exploration yielded the same segmentation as our custom software (Figure 8 right).

While the performance of this approach was quite good and compared very favorably with the cytopathology labs’ state-of-the-art [59,60,61], we introduced a few additional improvements: (1) by replacing our Euclidean square distances-based spectral segmentation with more advanced methods such as Support Vector Machines, we were able to get, reproducibly, 98%+ true positives in our analysis [65,66] even for specimens exhibiting clumps of cells, much better than current standards. This was described by an editorial in a special imaging issue of Science magazine ([67], quoted about 1000 times); (2) in order to automate the spectral image acquisition, thus greatly reducing cytopathologist workload, one could use a multimode-type microscope, with spectral imaging options. However, since spectral acquisition takes longer than intensity-based acquisition, one would prefer to effect such imaging only in selected areas of the specimen. Since cervical cancer cannot arise without the presence of HPV (the reverse not being true, of course), we designed a method whereby the HPV in specimens is labeled by a near-infrared probe, thus not affecting the visual appearance of the Pap smears. Upon imaging at low magnification and identifying the areas with HPV, one can restrict the spectral acquisition only to those areas. We patented this approach [68], and it is currently used in certain laboratories.

#### 2.4.2. Histopathology/Immunohistochemistry/Immunofluorescence

We took a similar approach to histopathology: since the dominant label (hematoxylin-eosin, known as H&E) is colorful but neither specific nor standardized, in spite of being around for 150 years, it was interesting to see whether spectral analysis (particularly user-guided segmentation yielding similarity mapping [61,62,63,64]) could contribute to our evaluation of specimens from real patients, especially as compared to more definitive tests such as immunohistochemistry (IHC) and immunofluorescence (IF). Some of our early results [59,60,61,62] are illustrated in Figure 9.

#### 2.4.3. Cell-Level Drug Candidate Analysis—Towards High Content Screening

As we started concentrating on spectral studies of live cells and tissues, we made the transition from slower spectral methods to much faster ones. The technology that had the most appeal was that of acousto-optic tunable filters (AOTFs), which have no moving parts, thus being able to change wavelength within microseconds, and are very versatile in controlling the light intensity and polarization as well. We researched their shortcomings as well and were able to improve their performance in both spatial and spectral resolution [69], through approaches we patented [70,71] and applied in building complete workstations for advanced applications [72,73]. In a collaboration with a pharma lab (Pfizer Research in Ann Arbor, MI, USA) we built a two-AOTF (excitation and emission, respectively, for spectral, temporal, and polarization control) workstation shown in Figure 10. The main purpose was to bring all the advantages of multimode microscopy that we used for toxicologic pathology (see above) to the in vitro, live-cell study of drug candidates. Such an approach is highly useful in identifying the best formulations to subsequently take into very expensive animal and human testing. We succeeded in thoroughly characterizing the cells of interest and their behavior (hepatic cells are shown in Figure 10, as affected by a proprietary drug candidate). We noted the exceptional heterogeneity of the cells’ response (see inset in Figure 10) by monitoring four to five important intracellular components (in the case shown nuclear morphology, intracellular calcium, mitochondrial membrane potential, and cell membrane permeability) [74,75].

#### 2.4.4. Intracellular Proteomics

Any tissue specimen obtained from a patient contains potentially a very large amount of specific information. The cell, as the basic unit of life, is also the locus of the derangements that ultimately cause cancer and should be the focus of studies aiming to understand mechanisms. Tumors arise as the result of the gradual accumulation of genetic changes in single cells, and identifying which genes encoded within the human genome can contribute to the development of cancer remains a challenge and a high priority in cancer research. Describing, evaluating, and quantifying the molecular alterations that distinguish any particular cancer cell from a normal one can predict the behavior of that cancer cell, as well as the responsiveness to treatment of that tumor in an individual. Understanding the profile of molecular changes in any particular cancer allows correlating its resulting phenotype with molecular events and should yield new targets and strategies for therapy.

Recent discoveries indicate that alterations in many of the cellular processes, pathways, or networks may contribute to the onset of cancer and could be used for therapeutic intervention. Therefore, it is important to put in place technologies that can detect molecular changes within the cell, without preconceived ideas about which information will be most valuable to monitor or what technologies will have the greatest impact. It is currently possible to study very specific changes in the expression and function of genes and gene products at the DNA, RNA, or protein level. However, many existing technologies do not adequately address specific issues, such as restrictions on the number of components studied in an experiment, limited cell number, sample heterogeneity, variability of specimen types, and cost-effectiveness. Innovation yielding novel technologies to study tumor specimens is needed. Recent advances in molecular genetics have made it possible to perform multiple correlated measurements on the cells of individual tumors and use such measurements to identify specific molecular subtypes of cancer and develop tumor subtype-specific combinations of targeted therapeutic agents.

The development and translation of new in vitro technologies for the multiplexed analysis of molecular species in clinical specimens requires a multidisciplinary approach. Progress in the application of prognostic factors in cancer will ultimately depend on the intelligent use of such factors in combination. The most robust combinations of such predictive factors are likely to be those based on relationships between tumor biology at the molecular level and clinical aggressiveness. A systematic hypothesis-testing approach for developing such combinations is both desirable and feasible and has been greatly facilitated by several recent developments. First, it has become increasingly apparent that there are specific patterns of molecular abnormalities that occur in individual tumors that are recapitulated in tumors from different patients [76,77], and that these patterns are of clinical prognostic value. These patterns provide useful starting points for the conceptual formulation of hypotheses regarding derangements in intracellular molecular network behavior and their effects on tumor aggressiveness. However, there are special difficulties in actually testing these hypotheses in studies on clinical samples, because the material available for study is limited in quantity, because there is extensive clonal heterogeneity within clinical tumor samples, and because of the limited degree to which fresh clinical material can be manipulated experimentally.

As gene products exert their function in a timely and spatially defined manner within a specific molecular and cellular environment, knowledge of qualitative in relation to quantitative gene expression profiles is central to the understanding of the role and activity of gene products in complex biological processes. We developed new imaging technologies [63,78,79,80] for the replacement of flow cytometry and laser scanning microscopy with more powerful approaches, allowing simultaneous imaging and quantitation of a large number of molecular species within the same cell. We have imaged 6–10 cancer-relevant proteins (including p53, HER2/neu, c-myc, ras, p21, cyclin D, etc.) in breast and lung cancer specimens from patients (see Figure 11).

This led us to multispectral imaging [64], which in our definition consists in using several spectral methods for imaging, simultaneously. It enables the optical probing and discrimination of intra- and extracellular molecules within tissues and provides a quantitative, dynamic picture of in situ and in vivo molecular interactions. Multispectral imaging has therefore major advantages in clinico-pathologic diagnosis and prognosis, and in therapy design and validation. The output is a digital, segmented, quantitated, and partially interpreted image of a fixed cell or tissue (on a microscopic slide). While in vitro gene expression and proteomic analysis provide only crude quantitative information of bulk tumor tissue, spectral imaging can generate quantitative biomarker information on a per-cell basis, and thus insights into complex interactions within the cellular pathways. This single-cell focus and subcellular resolution information is also extremely valuable in the assessment of the heterogeneous tumor composition in breast cancers as well as pulmonary carcinoma. Since multispectral imaging is amenable to automation and electronic data transfer and storage, it offers great improvement in time and cost reduction in the clinico-pathological routine to classify cancer disease processes through the analysis of a defined cluster of prognostic markers. Additionally, the effort in imaging a large number of intracellular components simultaneously is well spent: one six-color panel contains as much information as 30 four-color panels (the limit of current laser scanning cytometry), and one seven-color panel contains as much information as 210 four-color panels. As mentioned, our devices of choice in achieving the spectral and temporal imaging performance needed are acousto-optical tunable filters (AOTFs) [69,70,71,72,73]. Since the cells are studied in vitro, a large number of dyes and other markers (such as quantum dots) can be used, potentially raising the number of components that can be simultaneously imaged intracellularly to a record 18–20 (D.L. Farkas et al., in preparation).

### 2.5. Multispectral Multimode Microscopy

We found that adding lifetime, Raman, and coherent anti-Stokes Raman spectroscopies to our multispectral imaging arsenal [78,79,80] is technologically challenging but rewarding in what problems can be tackled with such new capabilities. We had success in a number of new areas, encompassing intracellular networks [81], mesoscopic studies [82] (see below, Section 3), angiogenesis [83,84], and stem cell differentiation [85], but only one example will be discussed here, monitoring gene expression in situ/in vivo, mostly because molecular optical imaging was new to the field at that time, and the results were very specific (see Figure 12): in cartilage, mutations in gene encoding filamin B disrupted vertebral segmentation, joint formation and skeletogenesis [86,87].

In asking what is the distribution pattern of filamin A vs. B along the growth plate and within cells, we were able to show that (a) filamin A, B (and C) participate in cytoskeleton organization and signal transduction; (b) mutations in the genes expressing filamins produce structural changes in the protein leading to numerous developmental anomalies in the brain, skeleton, and viscera; (c) filamins exist in vivo as dimers, and dimerization, leading to homo- and possibly heterodimer formation is mediated by interactions between carboxyl terminal sequences [86,87,88].

## 3. Pre-Clinical Optical Bioimaging


*“But the inadequacy of these microscopes, for the observation of any but the most minute bodies, and even those if part of a larger body, destroys their utility; for if the invention could be extended to greater bodies, or the minute part of greater bodies, so that...the latent minutiae and irregularities of liquids, urine, blood, wounds, and many other things could be rendered visible, the greatest advantage would, without doubt, be derived.”*
Francis Bacon [5].

We aimed to explore imaging with microscopic spatial resolution inside macroscopic objects (experimental animals), and we denoted this as mesoscopic imaging. Since, for best results, this imaging also needs to be in real-time, with a good temporal resolution, quantitative, and able to discriminate between different molecular species, a number of methods, biomarkers (including intrinsic ones), and analysis steps need to come together, in a multimode imaging setting, to achieve our goals. Our path to achieving this, and the advancement from animal models to human use, are reviewed below.

### 3.1. In Vivo Fluorescence Imaging of Cancer

#### Fluorochrome-Labeled Antibody Targeting for Molecular Imaging In Vivo

Tumor localization using fluorescence has been made practical by improvements in tumor targeting molecules, by the development of convenient near-infrared emitting fluorochromes, and by the availability of digital cameras having high sensitivity in this spectral region. Recent studies in animals have demonstrated that fluorochrome labeling of monoclonal antibodies confers adequate sensitivity and improved resolution. Simultaneous localization of multiple reagents is made possible by labeling with several different near-infrared emitting fluorochromes; thus, background subtraction and differential labeling of multiple tumor-associated components can be performed. Difficulties in using the fluorochrome labels are mainly related to light scattering and absorption in tissues, but detection of small tumors at depths of several millimeters is feasible. The major clinical use of this new technology is likely to be the endoscopic location of tumors. Scientific uses include studies of tumor metastasis, uptake and distribution of drugs and tumor-targeting molecules by tumors, and migration patterns of near-infrared labeled cells in vivo.

One of the promising areas of research is tumor visualization in vivo using fluorochrome-conjugated monoclonal antibodies. Relatively new cyanine conjugates that fluoresce in the near-infrared we developed offer improved resolution and high sensitivity for in situ studies of tumor growth and metastasis in animal models [89,90,91,92,93,94]. Near-infrared emitting fluorochromes have been developed that have high quantum yields, are easily conjugated to antibodies or other tumor-targeting agents, are visible through several millimeter thicknesses of tissue, and have the needed stability for labeling in vivo [93]. These fluorochromes are now commercially available. Moreover, high-resolution cooled CCD cameras that have adequate sensitivity in the near-infrared are now both available and affordable. Improved targeting agents, including second- and third-generation antibodies and antibody fragments, peptides selected from combinatorial libraries, and ligands for receptors that are altered, overexpressed, or selectively accessible in tumors, are emerging almost daily.

Fluorochromes have many advantages as in vivo labels. Each fluorochrome molecule can undergo many cycles of excitation and emission, yielding many photons, rather than giving rise to a single signal, as does radioactive decay. Light can be focused by lenses, allowing both a wide field for the collection of photons and good resolution. Red or near-infrared excitation light is used, not ionizing radiation. Differential targeting can be easily accomplished by using two or more separate fluorochromes, emitting at different wavelengths, as in differential isotope labeling. The first step in fluorochrome labeling in vivo was accomplished by Jain, whose group used fluorescein-conjugated monoclonal antibodies to study the details of antibody uptake in xenografted tumors implanted in rabbit ears, then in nude mice (see [95]). Fluorescein has the advantage of being readily visible to the naked eye; however, a significant background of tissue fluorescence occurs at the wavelengths used for fluorescein excitation and emission. Another problem with fluorescein is that its emission is sensitive to pH; emission is best above pH 8 and decreases sharply below pH 6. Thus, we would expect less than optimal fluorescence at blood pH, and intracellular fluorescein derivatives in acidic endosomes would be still less fluorescent. Furthermore, photobleaching is very significant.

Better fluorochromes have now been available for a while. Researchers at our Carnegie Mellon University center have synthesized a series of blue to far-red and near-infrared emitting fluorescent dyes, among them Cy2, Cy3, Cy5, Cy5.5, and Cy7 (see Figure 13 for structures). These fluorochromes are tailored for high quantum yield, good chemical stability, and ease of conjugation to various carriers and are now commercially available from GE. For Cy5, Cy5.5, and Cy7, both excitation and emission are at wavelengths where blood has relatively low absorbance. Fluorescence is insensitive to pH in the range from pH 4 to 9, and self-quench is low. Moreover, both scattering and background due to intrinsic tissue fluorescence are reduced by working at longer wavelengths, and the fluorochromes have high resistance to photobleaching. At high substitution levels, Cy5-antibody conjugates retain their brightness, in contrast to the quenching observed at high levels of fluorescein or rhodamine conjugation. The result is high sensitivity and a signal-to-noise ratio far superior to that of fluorescein, as assessed by fluorescence microscopy. The fluorochromes are stable in vivo in circulation. Erythrocytes surface-labeled using Cy3 and Cy5 were found to have a half-life of 40 days in rabbits with no diminution of fluorescence per cell. Because the fluorochromes have a convenient range of excitation and emission maxima, it is possible to visualize several fluorochromes in a given experiment by using multiple filters and image processing to remove spurious signals from spectral overlap; thus, background subtraction using a differentially labeled non-targeting antibody is possible [90,93]. Alternatively, multiple targeting reagents can be followed in a single experiment. The recent adaptation of acousto-optic tunable filter technology to deliver good-quality, high-resolution images makes visualization of multiple fluorochromes much more practical, as bandpass can be tailored to the needs of the individual experiment, and there is no need to use several different filters. Because of the poor sensitivity of the human eye in the far-red and near-infrared, the fluorochromes must be visualized using a camera. This is not a disadvantage, since electronic cameras have spectral sensitivity peaking in the near-infrared.

We have performed studies aimed at determining which of the currently available cyanine fluorochromes is most useful for in vivo tumor location. Direct comparison of four commercially available cyanine fluorochromes conjugated to the same monoclonal antibodies showed that Cy7 and Cy 5.5 were superior to the shorter wavelength dyes (Cy2, Cy3, and Cy5) for in vivo visualization of two different mouse model tumors [93,94]—see Figure 13 (left).

Scattering and absorption are the two key factors that diminish light flux significantly. In a homogenous, isotropic medium, Rayleigh scattering diminishes with the fourth power of wavelength; therefore, visualization in the infrared should be appreciably better than in visible light. In biological tissues, this simple wavelength dependence holds only qualitatively. We have taken a different approach to quantitate light penetration in biological tissues. We investigated the effects of tissue thickness on detectability and resolution by using multiple views of living animals labeled using fluorescent probes. Back-projection was used to reconstruct tomographic images of the fluorescence emission to better visualize fluorescent structures in living mice, to judge how well light at successively longer wavelengths penetrates. This approach provides a good method for assessing depth and resolution in a realistic in vivo model ([94] and Farkas, D.L., de la Iglesia, F., Galbraith, W. and Kanade, T., unpublished).

**Figure 13 molecules-26-06651-f013:**
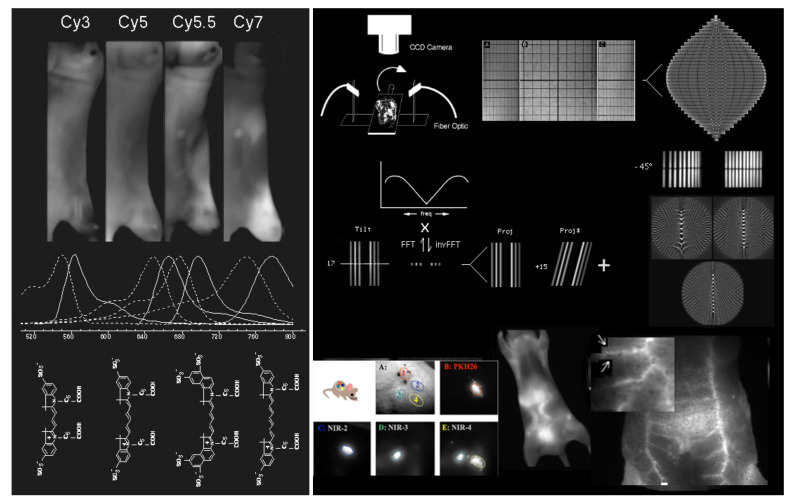
In vivo imaging of fluorescently targeted tumors: the importance of wavelength. Left: Tumors induced in a nude mouse were targeted with antibody, labeled with an equimolar mixture of Cy 3, Cy 5, Cy 5.5, and Cy 7. As the number of carbon atoms in the chains increases (bottom), so does the wavelength (excitation and emission) of the fluorophore (middle). Since the images (top) are of the same mouse, at the same time (but in different wavelength channels), one can see that the tumor (including a necrotic area) is better visualized the longer the wavelength. Cy 7 yields by far the best “transparency”, due to lowered scattering. Right: 3D imaging schematics illustrating our Digital Optical Goniometry (DOG) scans. Any features that can be highlighted in fluorescence (such as internal organs, blood vessels (shown), and targeted cancers can be mapped in 3D. Multiparameter fluorescence is also possible, as illustrated (insert, lower middle) with another family (PKH) of dyes [96] that can be wavelength-tuned (and we had access to several non-commercial ones).

Studies of model tumors in animals are simplified by using cyanine fluorochrome labeling. No radioisotopes are needed, so associated problems of radiation and disposal of radioactive tissue are avoided. Resolution is far better than that offered by conventional gamma cameras and superficial tumors, growth, and patterns of metastasis are easily investigated using fluorochromes [92]. Current results from targeting studies in vivo show that cyanine fluorochrome labels constitute a promising tool. The desirability of noninvasive, high-resolution imaging in living organisms, coupled with significant advances in component technologies, fluorophore synthesis, antibody purification, detectors, and digital image processing makes this an exciting and fast-evolving field. Major recent advances are based on better fluorescent antibody targeting, differential detection, and improved probes and optical design [95]

### 3.2. Multimode Imaging In Vivo

We were eager to extend our multimode imaging concept to preclinical studies of small animals. This had to be undertaken from the ground up, as there was no commercial offering to even constitute a platform for building on. We developed an advanced multimode optical imaging capability in a stand-alone system that enables functional mesoscopic imaging (whole-body or endoscopic imaging with microscopic resolution) of small animals in vivo and provides for quantitative, dynamic, and functional monitoring of chemo- and nanoconstruct therapy [97,98,99,100,101,102,103,104,105,106,107,108,109,110,111,112,113,114,115]. Many currently available imaging approaches have been applied to preclinical studies of cancer, stem cells, and pharmaceutical outcomes. Moreover, useful in vivo imaging may require several, preferably combined, and advanced imaging modalities to examine different but complementary characteristics of molecules, cells, or tissues. Although commercial systems perform well for standard imaging of small animals, they have limitations stemming from being single-modality instruments. Thus, a new multimode optical imaging system that is designed to be application-optimizable, with higher sensitivity and specificity has been required in order to overcome these limitations (Figure 14).

The instrument combines various in vivo imaging modes, including fluorescence intensity, spectral, lifetime, intra-vital confocal, two-photon excited fluorescence, and bioluminescence imaging. This makes it is a unique and comprehensive imaging platform for analyzing kinetic, quantitative, environmental, and other highly relevant information with macro- to microscopic resolution. This system can be optimized for various applications, and the combination of multiple imaging modes for increased contrast and complementary/synergetic information in chemotherapy assessment and cancer detection constituted our main emphasis.

We conducted nanoparticle [108,109] and theranostic nanoconstruct polycefin [113,114,115] evaluation in vivo and diagnostic and chemotherapy capability assessments with targeted constructs [108]. We have published extensively on this, and the details of the system (see Figure 14) are given in [107].

We will only discuss one representative application here (corrole chemotherapy assessment, [100,101,102,103,104,105,106,107,109,110,111]), and summarize the other applications in Table 1 (below).

The power of combining advanced optical imaging technologies can be brought to cancer detection and treatment assessment (including mechanistic studies) of HerGa, a novel single spontaneously self-assembled complex comprised of a cell penetration protein (targeting breast cancer cells that express Her2-neu) and a sulfonated corrole (part of a family of tunable dyes developed at Caltech by Harry Gray’s group). Basic optical characteristics (fluorescence lifetime and optimal two-photon excitation wavelength) were investigated first, as were fluorescence lifetime changes of HerGa due to cancer cell endocytosis, monitored in real-time. In addition, after adding corroles to breast cancer cells cultured in a temperature-controlled chamber, mitochondrial membrane potential variations in these cells, which indicate their health status, were monitored in order to investigate the mechanism of action of the corroles in cells. In addition, multimode optical imaging was performed to assess the capabilities of HerGa in breast cancer chemotherapy in vivo. Finally, the feasibility of multimode optical imaging of HerGa in cancer detection and delineation was examined.

Corroles have a variety of optical characteristics making them valuable as chemotherapy molecules. They emit very bright fluorescence with near-infrared wavelength. The fluorescence lifetimes of the corroles depend on pH and are less dependent on their concentration. Thus, the fluorescence lifetime imaging of the corroles could be utilized for the acquisition of functional and environmental information. Particularly, the temporal and spatial variations of the fluorescence lifetime of HerGa on cancer cells by endocytosis/drug effects were clearly shown in our studies [105,106,107]. We could confirm that untargeted S2Ga cannot be easily internalized into cells, but HerGa can be internalized through receptor-mediated endocytosis. The results also suggested that the internal environment of cancer cells is more acidic than cell membranes since the fluorescence lifetime of corroles is higher at low pH.

In multimode optical imaging in vivo for the assessment of HerGa, fluorescence intensity imaging allowed us to monitor dynamic accumulation of HerGa into a nude mouse in real-time. In particular, we observed that the HerGa was preferentially accumulated in breast tumors compared to S2Ga as judged by fluorescence intensity imaging [101]. Moreover, we monitored the accumulation kinetics of HerGa into the nude mouse at sequential time points using spectral imaging with ratiometric analysis. In the ratiometric spectral classification images, we could clearly distinguish between tumor and non-tumor regions due to more intense HerGa accumulation at the earlier time points after IV injection [105]. In the tumor regions, tumor vasculature actively arises through a process known as angiogenesis in order to provide blood to tumor tissues. Spectral imaging with ratiometric analysis enabled monitoring of the accumulation kinetics of HerGa more quantitatively and accurately than with typical spectral analysis methods. In addition, we could see that the HerGa was still present over the whole area 1 day after the injection. This spectral imaging with the ratiometric analysis method could also be useful for more accurate monitoring of the clearance of HerGa at later time points.

In addition, fluorescence lifetime imaging (FLIM) has been performed in order to acquire surrounding information (acidity) around tumor regions since it is known that the fluorescence lifetimes of HerGa at low pH are higher than that at high pH. The fluorescence lifetimes of HerGa accumulated in tumor regions are higher than in non-tumor regions and the histogram of fluorescence lifetimes in tumor regions had a greater population at 2.0 ns than in non-tumor regions. Finally, the multimode optical imaging of HerGa accumulation into specific organs and tumors, which were extracted from the same mouse after 4 days, also enabled the acquisition of different but complementary information, simultaneously. In the fluorescence intensity image of the organs, we could confirm the preferential accumulation of HerGa into the tumors compared with other organs. Importantly, in spectral classified images, HerGa was quantitatively discriminated from autofluorescence. While HerGa was clearly shown in whole tumors and in some parts of the liver, HerGa was very sparse in other organs (kidney, lung, heart, spleen, and muscle). Fluorescence lifetimes of HerGa in tumor regions are clearly higher than those in livers. From this result, we could infer that tumors have a more acidic environment than the liver and HerGa cannot be internalized in the liver. Finally, scanning as well as full-field two-photon excited fluorescence imaging of tumors, liver, and muscle provided high spatial resolution information, deep into the live tissue, clearly showing microstructures in the tumors and liver, but not muscle. Moreover, the fluorescence lifetime difference of HerGa in tumors and liver opens the possibility of HerGa being used for cancer detection and delineation. The combination of fluorescence intensity, spectral, ratiometric, and lifetime imaging enhanced contrast in cancer detection and delineation by HerGa for more accurate measurement. Two-photon excited fluorescence imaging of tumors provided additional highly resolved micro-structural detail. If transferred to a clinical setting, this may help physicians’ decision-making in cancer surgical intervention and detection without the need for rapid histopathological analysis of tissue biopsy during surgery.

Subsequent studies we conducted [105,106,107,108,110,111] showed that corroles in general, but targeted corroles, in particular, are near-ideal, versatile chemotherapy agents, as (a) they report on their status (location, concentration, surroundings) by having intrinsic, intense, and wavelength-tunable fluorescence that is sensitive to their environment (through FLIM); (b) they can delineate the cancers of interest in vivo, by fast and preferential targeting and accumulation, especially as assessed by ratio-metric fluorescence spectral imaging; (c) they can preferentially kill the targeted cancer cells in a number of (additive, even synergetic) different ways, based on their chemistry and additional photo-induced effects such as singlet oxygen-mediated toxicity affecting cells and the mitochondria within them [110,111].

### 3.3. Intrinsic Imaging without Contrast Agents

With proper experimental planning, light originating in a sample (including a living body) is both target-specific and content-rich. Semantic optical imaging is thus enabled, allowing attribution of instant significance to a signal. The often used (but sometimes misused) term optical biopsy captures this exciting concept, in the sense that an entity or feature of interest could be detected, located, and quantitated within the body, noninvasively, with diagnostic power approximating that of a traditional biopsy. We have come to believe that a major obstacle in moving optical imaging diagnostic technologies from the laboratory to the bedside is that the heavy use of contrast agents in research cannot be duplicated intrasurgically. Although imparting higher specificity and signal-to-noise to imaging, agents routinely used in live-cell, fixed tissue, and animal experiments are not FDA-approved and thus not transferable for use in humans. This is not likely to change soon and underscores the need to concentrate on optical imaging requiring no extrinsic contrast agents. We aimed to test these methods on unstained cancer specimens, ex vivo and in vivo.

A key issue in comparing an in vivo test with the pathology “gold standard” is that the chemical processes applied by the pathologist to the specimen distinctly alter it. Often, when the processing is complete, the specimen has been deformed or warped and toxic contrast agents have been added. We chose to address this issue through the use of Odd-Even slicing of the sample; this involves having the pathologist prepare the embedded tissue sample and then slice it using the micro-keratome. Then, the rest of the traditional H&E staining procedure is completed using only the odd-numbered slices. The even-number slices are instead placed on microscopy slides for unstained spectral imaging, with the paraffin removed from the slide. The unstained (even) slides were then imaged using reflected light, and we compared spectral imaging on unstained slides (even) to their corresponding stained slide (odd) that exhibits the same topology, to verify our results against the gold standard, as defined by the pathologist. We settled on the use of ink as a fiducial marker. The remainder of this experiment had two phases:

For the first phase, nine pairs of stained-unstained slides were obtained and classified. From these preliminary results, we were able to segment regions of breast cancer from regions of normal tissue using the minimum squared error classification scheme. The segmentation results of the stained slide are very similar to the results of the corresponding unstained slides of breast cancer. For the second phase, we progressed to imaging fresh unstained ex vivo specimens, the images from which the library was defined with separate images of normal breast tissue, normal breast tissue stained with ink, and pure breast cancer from a 10-day old tumor. A total of 22 animals were induced with cancer, and more than 2000 images were taken. We experimented with various classification schemes and normalization techniques and ultimately arrived at using a normalized Dimensional Analysis with Mahalanobis distance. We calculated the sensitivity, specificity, positive predictive value (PPV), and negative predictive value (NPV) using generally accepted clinical calculations to be 96%, 92%, 95%, 95%, respectively. Our results [116,117,118,119,120,121] demonstrate that spectral imaging and classification have the potential to detect cancer regions within fresh unstained breast tissue.

The results summarized illustrate our ability to achieve quality pathologic assessment of unstained tissue. The approach likely to yield the best results (including margin assessment, while cancer surgery is still going on) will require combining spectral imaging with one or more other optical methods, similar to the multimode studies described above for animal models. The success of any such undertaking will be dependent not only on regulatory approval but also on an adoption willingness by the practicing surgeons. An important element in planning this should be a very informative display of the multimode images obtained, in real-time, but also some level of on-the-fly interpretation, such as a separation of entities of interest (by e.g., pseudocolor or dual display) so that everything of primary interest can be located and quantitated when needed in the procedure.

In order to address this, we developed an intelligent spectral signature-based analysis and display scheme [64,119] that is illustrated in Figure 15. In live preclinical imaging, we could distinguish the various component tissues by location and spectral signature, and display them in separate channels (bottom), but could also reconstitute a realistic topological representation of the live tissue area, with a hue-saturation-intensity (HSI)-based display (top, right) that is more informative and realistic than a straightforward spectral segmentation (top, second from left).

### 3.4. Coherence-Based Imaging

Optical Coherence Tomography (OCT), this elegant imaging method invented at MIT [122], uses coherence properties of light to detect the photons of interest for imaging from a background that could be 6–7 orders of magnitude higher. Similar in concept to ultrasound imaging but using (near) infrared light, it allows better penetration (2–3 mm) into tissue than conventional optical imaging and has reached the speed (~video rate or higher) and spatial resolution to allow it to produce highly useful images of cells, in vivo, with no contrast agents, in real-time. By now, there are a large number of commercial offerings, allowing good access to both the research and clinical communities to this great new tool that has progressed to being useful in the clinic.

We developed our own high fidelity OCT system [123,124,125] with dual-wavelength, high resolution, and superior dynamic range (~140dB), and we examined its utility in imaging skin [125] and urinary bladders. We systematically studied the morphological alterations associated with diseases of the bladder including urothelial tumors, in an animal model closely approximating the human case [126]. The methodology, incorporating and comparing OCT with cystoscopy (by surface imaging) and excisional biopsy (by histological evaluation), allowed the evaluation of OCT’s potential for the noninvasive diagnosis of transitional cell cancers (Figure 16).

Bladder cancer is the fifth most common cancer in the US. If detected and excised prior to invasion or metastasis, bladder carcinoma is curable. However, none of the current detection methods, e.g., urine cytology, intravenous pyelogram, MRI, and ultrasound provide sufficient sensitivity or specificity in predicting the prognosis of early bladder cancers and staging their invasions because of resolution limitations or technical imperfections of these methods. In a methyl-nitroso-urea (MNU) instillation-induced cancerization model, our experiments using the OCT system we built showed [126] that the micromorphology of the porcine bladder such as the urothelium, submucosa, and muscles is identified by OCT and well correlated with the histological evaluations. OCT detected edema, inflammatory infiltrates, and submucosal blood congestion as well as the abnormal growth of urothelium (e.g., papillary hyperplasia and carcinomas). By contrast, surface imaging, which resembles cystoscopy, provided far less sensitivity and resolution than OCT, and histology showed obvious changes in tissue during cancerization, but also additional changes (vs. OCT) that were due to tissue preparation for histopathology. This was the first OCT study of any tumor documented in a systematic fashion throughout its entire evolution, and the results suggested the potential of OCT for the noninvasive diagnosis of both bladder inflammatory lesions and early urothelial abnormalities and cancers, which conventional cystoscopy often misses, by imaging characterization of the increases in urothelial thickening and backscattering. More details are given in Figure 16 and its legend.

## 4. Functional Imaging


*“Science is driven by ideas, but paradigm shifts are often a direct result of advances in technology.”*
Peter C. Doherty (Nobel laureate)

### 4.1. Intrasurgical Topological Guidance: Hirschsprung’s Disease

Hirschsprung’s disease is the congenital absence of specialized nerve cells (ganglion cells) primarily affecting the lower portion of the colon. When Hirschsprung’s disease is untreated, it causes severe constipation that can lead to massive dilatation of the colon (megacolon), colonic obstruction, often leading to death by overwhelming infection. It affects approximately 1 in 5000 live births. Nowadays, approximately 90% of Hirschsprung’s patients are diagnosed in infancy and undergo surgical therapy very early in life. We highlight this application of advanced optical imaging as (a) it was relatively easy to perform the experiments in vivo, and the results were convincing, (b) it required no biomarkers, staining or other intervention, relying on intrinsic signatures, and (c) because of this, transitioning this approach to the pediatric clinic seems straightforward and desirable.

The current surgical therapy for Hirschsprung’s disease is to perform a minimally invasive pull-through procedure in the first month of life. The initial step is to identify the level at which ganglion cells are present in the colon. This is routinely performed by laparoscopically procuring multiple small biopsies of the colon wall which are then sent for rapid frozen sections to pathology. The pathologist looks for the presence or absence of ganglion cells (and other features) to determine the diagnosis in each specimen, thereby determining the level where the transition from normal to the aganglionic colon occurs. The precise level of the transition zone is absolutely critical to performing the next stage of the surgery. The aganglionic colon is then removed and the normal colon is pulled through the anus and sutured in place. This surgery avoids the need for a colostomy and has proven to be safe and effective. One of the critical portions of this procedure is the accurate and precise determination of the transition from normal to aganglionic colon. This portion of the surgery typically takes 45–60 min, during which time no operating is performed. Most of the time is absorbed by waiting for the frozen section information with the patient remaining under general anesthesia. The cost of the operating room time is significant and waiting for frozen sections can be costly during this period of relative inactivity.

Despite the expertise of pathologists reading rapid frozen sections, they are not 100% accurate in determining the level of aganglionosis. On occasion, albeit uncommon, the level is not accurately determined, and a patient might have more or less colon removed than is appropriate. If too little colon is removed, thereby leaving aganglionic colon, the patient would likely develop significant constipation. This scenario may potentially require additional surgery to remove more of the colon. Conversely, if too much colon is removed, the patient would have increased stool frequency (diarrhea) which can result in body salt and mineral imbalances, dehydration, and skin breakdown.

Preclinical studies using similar spectral imaging techniques were performed on a mouse model of Hirschsprung’s Disease [127,128,129]. The device sampled images from normal and aganglionic colon in these animals. Analysis of the spectral signature images showed a clear distinction between the normal and aganglionic colon when correlated with pathological analysis. We developed an algorithm that could distinguish normal from aganglionic colon with Sensitivity = 97%, Specificity = 94%, Positive Predictive Value = 92%, Negative Predictive Value = 98% [127,128]. These studies showed “proof of concept” that spectral imaging techniques could be used during the course of operations to help surgeons distinguish normal from diseased tissue without requiring biopsy.

Our goal is to have this same spectral imaging technology and analysis applied to children with Hirschsprung’s Disease. This study would generate the spectral imaging data with precise pathologic correlation to develop an intraoperative tool to distinguish normal from aganglionic colon in real-time, more accurately, less invasively than what is currently available. With the reduced time of the operation and no requirement for intraoperative pathology consultation, this technique has the potential to reduce the cost of treatment as well. Figure 17 shows some of the data and a graphic summary of the results.

### 4.2. Imaging Stem Cells In Vivo

Stem cells are a wonderful discovery of modern medical science, and likely to massively change certain areas of it. The enthusiasm about what is possible in regenerative medicine is intense and probably justified. Nevertheless, given the complexity of everything involved, we need to be cautious: in current scenarios (certainly in the imagination of the public), we just add (inject?) these wonderful cells into the body, and (a) everything that we hope for happens and (b) nothing that is unexpected or indeed negative happens. Even if this were the case (but sadly it is not), for regulatory and, frankly, self-assurance reasons, would it not be nice to see (test by visualization) what happens to these cells inside the body, in time.

We had an early interest in imaging these cells (even before they were called stem cells) [130]. We developed an in vivo imaging approach to monitoring hematopoietic stem cells (HSCs) in a mouse model and used it to access the dynamics of stem cell fate in the bone marrow. We were able to track the cells by labeling them with a family of dyes called PKH [96] that intercalate into the stem cell membranes in a very stable way, lighting up the cells by fluorescence that allows imaging through a bone window system we developed. There are several dyes in this family available, with different excitation/emission wavelengths options, so that multiple cell types can be tracked simultaneously, or certain “tricks” can be deployed, such as fluorescence resonance energy transfer (FRET) that allowed us to distinguish live cells from dead ones, weeks into the experiment, still in vivo. Figure 18 illustrates some of these findings and shows (right) the mechanistic model derived.

We have published extensively on this [131,132,133,134,135,136,137,138,139,140,141,142,143,144], with [139] a brief review of the most interesting findings. We could show that the HSCs were alive and functional after three weeks, having undergone a cycle of homing, adhesion, cluster formation, colonization, engraftment (and subsequent differentiation), just as predicted and hoped for, but with more details and mechanistic clues. Other investigators have started using these imaging approaches with promising results [145]. We believe that optical imaging should become a very useful tool in this field, greatly aiding in bringing the predicted (promised) regenerative wonders to the clinic.

**Figure 18 molecules-26-06651-f018:**
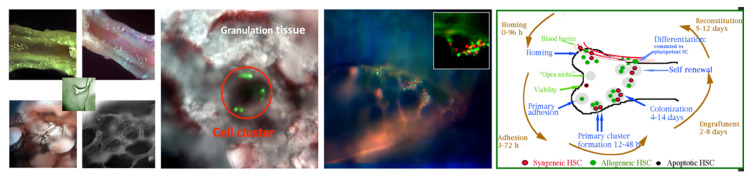
Imaging hematopoietic cell engraftment and survival in the recipient bone marrow. Far left: Intravital microscopy in a bone window. In vivo BM in reflected light (left) and UV-excited autofluorescence (right), at different magnifications reveals resolution, trabecular structure, good survival. Middle left: Formation of a primary cellular cluster. In vivo BM imaging 3 days after I.V. injection of syngeneic BMC in busulfan-conditioned B10 mice. Superposition of brightfield and fluorescence. (20x). Middle right: FRET imaging viability assay. After 20 days in vivo, stem cells engrafted into the marrow were imaged by Fluorescence Resonance Energy Transfer to distinguish live cells (green) from dead (red), at 10x magnification. Far right: Topologic/mechanistic steps (homing, adhesion, engraftment, reconstitution) that can be inferred from imaging HSCs in vivo [138].

### 4.3. Neuroimaging Primary Events: Calcium Transients at the Neuromuscular Junction

We developed an interest in calcium measurements in preclinical systems and realized the advances possible using advanced optical methods [146,147,148,149]. Primary events, such as action potentials, depend heavily on calcium dynamics and signaling. The nature of presynaptic calcium (Ca^2+^) signals that initiate neurotransmitter release makes these signals difficult to study, in part because of the small size of specialized active zones within most nerve terminals. Using the frog motor nerve terminal, which contains especially large active zones, we showed that increases in intracellular Ca^2+^ concentration within 1 ms of action potential invasion are attributable to Ca^2+^ entry through N-type Ca^2+^ channels and are not uniformly distributed throughout active zone regions. Furthermore, changes in the location and magnitude of Ca^2+^ signals recorded before and after experimental manipulations support the hypothesis that there is a remarkably low probability of a single Ca^2+^ channel opening within an active zone after an action potential. The trial-to-trial variability observed in the spatial distribution of presynaptic Ca^2+^ entry also supports this conclusion, which differs from those of previous work in other synapses.

The main reason that we undertook this study was the realization that simultaneously high spatial, spectral, and temporal resolution imaging was absent from the field of neuroscience. In particular, while electrophysiological measurements were quite well resolved temporally, the de-facto fast imaging standard was video rate. It was known that primary events (such as calcium transients) are much faster than that, and the two-AOTF microscope workstation we built seemed suitable to investigate the possibilities (inside a Faraday cage), with its 275 nm in-plane spatial resolution, 50 µs temporal resolution, and ~8 nm spectral resolution. To record single (electric stimulation-induced) events with sufficient S/N, we used a back-thinned, cryogenically cooled CCD camera (Roper). The setup and some results are shown in Figure 19. One can notice that a change of even 1 ms in stimulation time, or 12 ms in delay (to imaging) yields completely different patterns and intensities of Ca^2+^ concentration, underscoring the need for the outstanding temporal resolution in such studies.

Specifically, it was known that on each stimulation, approximately 1/100 of the vesicles are activated. The question is, is this due to a low probability of calcium channel opening, or to a low probability of vesicle release? We could answer such questions by imaging the entire terminus immediately after a single stimulating pulse. We used a fast “snapshot” approach to imaging Ca2+entry into adult frog motor nerve terminals and studied the spatial distribution of Ca2+entry during single action potential stimuli at low frequency. Using this method, we observed spatially isolated sites of Ca2+entry in active zone regions of the nerve terminal that disperse with time after action potential invasion and vary in their location with repeated trials. We estimated that between 6 and 50 calcium channels might be under each pixel that samples a portion of a single active zone. If we considered the range of estimates for the number of calcium channels that might be sampled by a single pixel and the range of possibilities for the likelihood that an N-type calcium channel will open during a single action potential stimulus, we could determine whether we would expect to observe a graded calcium signal or one that includes a significant number of failures during low-frequency action potential stimuli. If there are many calcium channels (~50) sampled by each of our pixels, and a relatively high probability (greater than~0.2) for calcium channel opening during an action potential stimulus, our calculations predicted a calcium signal that is little changed with repeated trials and shows a graded change in intensity after partial blockade of calcium channels.

This was not what we observed. In contrast, if there are few calcium channels sampled by each pixel (approximately six) and a relatively low probability (less than ~0.2) for calcium channel opening during an action potential stimulus, our calculations predicted a calcium signal showing large variability with repeated trials, some increase in the number of trials in which there is a failure to detect calcium signal after the partial blockade, and no change in the intensity of signal detected at entry sites that remain unblocked. Our data were highly consistent with this prediction and opposite to what was known.

### 4.4. Oxygenation Mapping

The brain needs oxygen for functioning, and a detailed understanding of the complex topologies and mechanisms involved, in sickness and health, requires quantitative, spatially resolved, and preferably non-invasive methods of investigation. It became clear that the AOTF-based microscope we developed (see Section 2.4.3 and Section 4.3 above, and [73]) is ideally suited for such purposes.

We decided to concentrate on an animal model (brains of living mice) where we mapped, simultaneously, the oxygen saturation of hemoglobin (sO_2_) using spectral imaging, and the oxygen tension (pO_2_), using the phosphorescence of probes developed to assess the concentration of oxygen—their only quencher under our circumstances [150,151]. Spectral signatures are most informative of hemoglobin oxygen saturation between 500–600 nm, with the absorbance (measured in reflectance mode here) changing shape in a characteristic and known way. The phosphorescence lifetime of the particular palladium-porphyrin derivative we used changes from 600 µs under full oxygenation to 2 ms in the absence of oxygen, and we used this feature to deduce oxygen concentrations from the changes we measured in a lifetime, in frequency-domain measurements. The time resolution of AOTFs is sufficient to access this domain, and the principles behind our approach are outlined in Figure 20 (right).

We found that we could map, with high spatial and temporal resolution, in about 1 s, the sO_2_ and pO_2_ in the brain of a living, breathing mouse (Figure 20 left and middle). These types of maps have not been achieved before, and with the right technologies, they became relatively easy to obtain, with interesting future applications in assessing the role of oxygen and hemoglobin during functional activation of neuronal tissues.

In order to show that this was fully functional imaging, we could elicit changes in these parameters and could document the outcomes quantitatively. As the inspiratory oxygen was stepped from hypoxia (10% O_2_) through normoxia (21% O_2_) to hyperoxia, measured sO_2_ and pO_2_ levels rose predictably and reproducibly. A plot of one against the other in different arterial and venal regions of pial vessels conformed to the known sigmoidal shape of the oxygen-hemoglobin dissociation curve, providing further validation to our mapping. Interestingly, these curves, while still sigmoidal, were quite different between different mouse strains (see bottom of Figure 20), something that was unknown and surprising to the research community (F. de la Iglesia, personal communication).

Not being in position to show the mice images on a screen, as carried out in some functional MRI-based oxygenation mapping in humans, in a further test of functional response we used amphetamine stimulation in the living mice and measured their sO_2_ and pO_2_ changes. We found that in most cases both these quantities decreased upon amphetamine administration, a rather unexpected result (see Figure 20 bottom) that could only be accessed using the new technologic approach described here. The very same imaging technology was used in subsequent oxygen tension imaging in the retina [152,153].

Overall, the spatiotemporal resolution and cost of these measurements compare very favorably with those obtainable by functional MRI, which for pO_2_ remain indirect and based on the assumption that the sigmoidal sO_2_ vs. pO_2_ curves never change.

## 5. Clinical Photonic Imaging


*“Progress in science depends on new techniques, new discoveries and new ideas, probably in that order.”*
Sydney Brenner, Nobel laureate

Our ultimate goal, and one that we should never lose from sight, is to bring advanced technologies in optical bioimaging to the clinic, in ways that make a positive difference. Therefore, the work described up to now in this review was, in a way, preparatory for what could be adapted, fine-tuned, and deployed in a setting such as a diagnostic, procedure, or operating room (OR), and we have denoted this accordingly in the text, unless the difficulty of the task requires a different combination of methods, or even entirely new ideas and tools to be brought to bear. Naturally, since with all its advantages documented above light still has one disadvantage in human use, namely its relatively modest tissue depth penetration, certain clinical targets are better suited for optical investigation than others. Notable amongst these are the skin and the eye, optically accessible in totally non-invasive ways, and the respiratory and gastrointestinal tract, accessible through endoscopy. We will concentrate mostly on these here, not forgetting that any methods developed have to fit with existing and feasible clinical procedures in an OR that is capable of accommodating the new solutions [154].

### 5.1. Neurodegeneration Imaging

Alzheimer’s Disease (AD) is a devastating neurodegenerative condition that currently has no treatment. According to a recent review [155] 5.5 million Americans have Alzheimer’s dementia. By mid-century, the number of people living with AD in the US is projected to grow to 13.8 million, fueled in large part by the aging baby boom generation. Since AD-related expenditures today are already in excess of USD 259 billion/year, this would be not only terrible for those affected, but also ruinous economically. Today, someone in the country develops AD every 66 s. By 2050, this would change to every 33 s, resulting in nearly 1 million new cases per year. AD, even underreported as it is, has become the sixth leading cause of death in the US and the fifth leading cause of death in Americans age > 65 years. Between 2000 and 2014, deaths resulting from HIV, stroke, heart disease, and prostate cancer decreased 54%, 21%, 14%, and 9%, respectively, whereas deaths from AD increased 89%, making AD the health crisis of our generation. Naturally, this yields extraordinary human interest (medical, social, and business) in the improved detection and treatment of AD, and there seems to be general agreement that the one development that could change this dire picture is much earlier detection of AD. We proposed a new approach to this challenge, based on high-resolution and specificity, non-invasive multimode optical imaging of biomarkers the retina, in living patients, for very early AD detection. We believe the same quantitative method could also be subsequently used for treatment assessment and drug discovery.

*Why optical imaging*? As outlined above, the biomedical applications of optical imaging are extremely broad, due to their intrinsically high sensitivity and specificity for discriminating between diseased and non-diseased tissue in a real-time and non-destructive manner, in vivo, and because the advanced technologies involved are both mobile and affordable, having been originally developed for space, defense, and mass-market applications. Hyperspectral imaging is one such optical method, and we have used it extensively in cellular, molecular, and clinical investigations.

*Why target the retina for AD research?* Noninvasive monitoring of β-amyloid (Aβ) plaques, the neuropathological hallmarks of Alzheimer’s disease (AD), is critical for AD diagnosis and prognosis. There is universal agreement on the fact that these plaques are early biomarkers of AD and somewhat less universal agreement on whether they are causative in AD. Visualization of Aβ plaques in the brains of live patients and animal models is limited in specificity and certainly in resolution. The retina, as an extension of the brain, presents an appealing target for a live, noninvasive optical imaging of AD if disease pathology is manifested there. The eye is optically accessible in all its components, making it a window into the workings of the brain. We were interested in Alzheimer’s Disease studies [155,156], and this author hypothesized [157,158,159] that the universally recognized primary biomarker of AD, amyloid-beta plaques, could be visualized in the retina early, in order to monitor primary events on the path to AD. We have shown this was possible, using rather makeshift laboratory equipment, capable of detecting low fluorescence with high spatial resolution.

We started with animal models (mutant mice developing AD fast) and found [157] that in transgenic mice but not in wild type, retinal Aβ plaques were detected following systemic administration of curcumin, a safe plaque-labeling fluorochrome that crosses the blood-brain and blood-retina barrier and attaches to the plaques. Moreover, retinal plaques were detectable earlier than in the brain and accumulated with disease progression, paralleling events in the brains. An immune-based therapy effective in reducing brain plaques significantly reduced retinal Aβ plaque burden in immunized versus non-immunized AD mice. In live AD mice, systemic administration of curcumin allowed—for the first time—noninvasive optical imaging of retinal Aβ plaques in vivo with high resolution and specificity, while plaques were undetectable in non-Tg wt mice (see Figure 21, right). Our discovery of Aβ specific plaques in retinas from AD patients, and the ability to noninvasively detect individual retinal plaques in live AD mice establish the basis for developing high-resolution optical imaging for early AD diagnosis, prognosis assessment, and response to therapies. We also identified retinal Aβ plaques in postmortem eyes from AD patients and in suspected early-stage cases, consistent with brain pathology and clinical reports; plaques were undetectable in age-matched non-AD individuals [157]. These studies were followed up [160] with convincing human trials, and (see references therein) a whole new field of research emerged using this retinal imaging approach.

Since in our view the purpose of a review is not only to enumerate or categorize past work, but also to draw methodological conclusions leading to future strategies, we point out that when it comes to biomedical research and intervention, we believe that

You cannot effectively fight what you do not understand.You cannot properly understand what you cannot visualize.Best way to visualize is by imaging, with high spatio-temporal discriminationThus, the recipe is that we need to image
The right thing (primary pathology/biomarkers)In the right organism (humans, not mice)At the right time (as early as possible in the disease evolution)In the right place (best area to look, even if not prevalent, e.g., retina for AD)The right way (i) non-invasively; (ii) with high resolution (spatial, temporal); (iii) dynamically/repetitively; (iv) intrinsically (without added contrast agent); (v) sensitively; (vi) vs. other biomarkers (e.g., vasculature, in the case of AD), imaged simultaneously; (vii) with depth penetration.


Based on an examination of the relevant literature, we noticed that for AD,

–more than half of all academic research focuses on non-primary pathologies (see 4a above)–most of academic research is in animal models (4b)–all clinical research is in late-stage disease (4c)–most of both academic and clinical research is in the brain (4d)–therefore, spatio-temporal resolution is poor (4e.ii)–all imaging is contrast agent-based (4e.iv) (but still better than subjective evaluations)


*All of these trends appear very inefficient in addressing AD.*


In our original new AD approach [157], we focused on

–amyloid plaques (primary pathology) (4a), using curcumin (biomarker) fluorescence–in both mice and humans, translationally (4b)–early in the disease (in mice) (4c)–in the retina, in both animals and humans (4d)–non-invasively, with high resolution, dynamically, and repetitively (4e.i–iii)


*We believe that all these were innovative, and major improvements over existing approaches.*


However, the bioavailability of the exogenous fluorescence probe used (curcumin) is quite variable, both temporally and between subjects, and this could affect quantitative evaluations. Thus, there were further improvements we estimated still needed:–imaging more of the retina (by larger angle scanning) (4d)–imaging without contrast agent (intrinsic signal: autofluorescence) (4e.iv)–imaging with higher sensitivity (for earlier detection) (4e.v)–imaging hyperspectrally (for better discrimination, and vs. e.g., vasculature) (4e.vi)

To implement these, we are developing a novel, multimode confocal scanning laser ophthalmoscope (MM-CSLO) of our own design, as we believe that (a) only a CSLO can properly image the biomarkers of interest with the proper spatial resolution, and (b) only the new MM-CSLO of our own design will have the enhanced sensitivity (needed to detect intrinsic signals), scan angle and versatility to address the ambitious goals of proper non-invasive optical imaging for AD.

Current retinal screening techniques make it difficult to accurately assess AD progression. Advantages of using advanced (multimode, adaptive optics) CSLO over traditional fundus photography include improved patient comfort through less light exposure for the eye, and non-mydriatic methods. The further importance of using MM-CSLO over optical coherence tomography (OCT) is the better lateral resolution, and the simultaneous measurement of anatomical (cell detection and microvasculature) and functional/biomarker (by fluorescence) endpoints, with high specificity that is lacking in OCT. Furthermore, our quad-galvo confocal scanning imaging system enhanced by adaptive optics (AO) provides 3D high-resolution images of the retina over a wide field of view with cellular resolution (3–5 μm with AO) to detect retinal vascular cells and small (~µm) aggregates of biomarker, in this case, amyloid-beta. Further details will be given elsewhere (A.G. Nowatzyk and D.L. Farkas, in preparation).

Overall, ours is a wide field of view in vivo cellular imaging instrument based on CSLO that could become an effective tool for routine retina assessment in the clinics and also applicable to a major current medical challenge, AD, in a new way: by focusing on very early diagnosis based on an intrinsic biomarker (amyloid-beta autofluorescence). The specific novel aspects (see Figure 22) are:–A new, laser-based hyperspectral excitation, allowing unique image acquisition–A new [161], quad-galvo technology in CSLO scanning, allowing flexible positioning of the beam’s pivot point at the eye’s pupil, providing a wider field of view than existing systems.–A new [162] photon detection technology, based on an unusual amplification scheme –3D imaging capability over the entire thickness of the retina with cellular resolution is provided. Confocal imaging provides high spatial resolution, and because complicated scan lenses are not required in the quad-galvo design, the optical system is relatively simple but has higher NA.–Simultaneous (multimode) measurement of oxygenation, microvasculature structure, and plaque autofluorescence with our MM-CSLO should allow assessing treatment-induced changes in these and thus provide new insight into AD progression and even etiology.–Investigation of the chronological order of neurovascular disfunction and AD. For example, it is still not known whether there is a correlation between vascular damage/oxygenation changes and AD, as some hypothesized. We believe we can address this in new ways.

All this should help to quantitatively characterize the early events and hopefully mechanisms of AD. Topologically quantified biomarkers that have been identified as playing a role in AD, even if the causal nature of this role has not been determined, can greatly help shed light on possible mechanisms of AD onset and early progression. This approach also has the potential to identify a new therapeutic strategy with a high likelihood of clinical translation, or even allow re-visit past (drug candidate) failures, as these were tested in stages of AD that were too advanced and thus unlikely to respond to treatment. Finally, since in most remaining unmet clinical challenges (such as HIV) a “cocktail” of drugs has been found useful, the imaging support allowed by MM-CSLO would enable testing and fine-tuning such drug or treatment combinations for AD, in later applications of our multimode imaging method.

### 5.2. Dermatological Imaging

Studying skin is a particularly good application of biomedical optical imaging. Only one specific example will be discussed here, due to its overall importance in potentially saving lives.

Melanoma is an increasingly lethal form of skin cancer, especially when detected in later stages. Melanoma risk during a lifetime has increased 30-fold since 1935 and is still the fastest growing cancer both in the U.S. and worldwide. About 80,000 patients will be diagnosed with melanoma of the skin this year and about 10,000, or more than one patient per hour will die. Survival rates strongly favor early diagnosis, ranging from 99% for early, primary site detection to at best 15% for late or metastasized detection. Approximately USD 2.4 billion is spent in the United States each year on melanoma treatment. Treatment costs average USD 1800 for early and USD 180,000 for late detection. This indicates significant cost savings by diagnosing melanoma earlier [163]. Despite great effort worldwide, no significant advancements in treatment have occurred, and thus early detection is by far the most effective means of fighting this disease that accounts for 75% of all skin cancer deaths.

We believe that any aspect of melanoma that can be studied with simple, commercial tools (such as a dermoscope) is easier to document, quantitate, dynamically follow, and understand when more advanced methods, such as hyperspectral and multimode imaging are deployed. We began our investigations in academia, collaborating with the Univ. of Pittsburgh Melanoma Center, by applying spectral imaging to gene expression analysis [164], tumor growth blocking by induced apoptosis [165], upstream regulators, and downstream targets of STAT3 in melanoma precursor lesions, before and after interferon treatment [166], and eventually built our first multimode (spectral, polarization control and autofluorescence) imaging dermoscopy workstation to investigate suspicious moles before removal and histopathology in patients at high risk for melanoma [167], where the instrument proved extremely useful (100% sensitivity and specificity in a small patient study [167]).

This line of investigation was continued in a small startup [168,169,170,171,172,173] and culminated in a second-generation SkinSpect multimode dermoscope (Figure 23) with much better performance (50 visible and NIR wavelengths at 10 nm resolution, variable polarization control, separate autofluorescence channel, faster acquisition at ~4 s, much better analysis software, with parallelization-accelerated processing, etc.) being deployed in clinical studies (40 and 100 patients, respectively) at two very prestigious medical centers, where our approach and results could be compared not only to standard histopathology, but also other dermatological imaging methods, such as reflectance confocal microscopy and spatial frequency domain spectroscopy [174,175,176,177,178]. Our results were very promising, including the ability to assess Breslow thickness non-invasively in living patients, in seconds, and will be readied for publication (in preparation). Regulatory approval for use was obtained in Canada (so far).

Considering the need for less complicated, less expensive, and mobile devices that go beyond what a dermoscope provides, and emulating some of the main features of our SkinSpect, we put some efforts into developing MobileSpect, a smartphone-based dermoscope for self-imaging (not self-diagnosis), in the hope that with teledermatology/telepathology help these could become a new standard for everyday use by both clinicians and patients, helping with the critically important screening for suspicious lesions [179,180,181,182,183], and found the device useful in other dermatologic applications, such as psoriasis [184]. This work continues, with other applications such as rosacea, wounds, bedsores, burns, and even cosmetic uses coming into focus. We protected the IP [185,186], reviewed the spectral component of our work [187], and extended the approach to other fields such as food quality and safety [188] that could benefit from a very similar set of technologies.

### 5.3. Endoscopic Imaging

#### 5.3.1. Hyperspectral Imaging of Mucosal Surfaces in Patients

We start the review of endoscopic imaging with this application because within a couple of years it evolved from practically non-existent to relatively widespread, while rising in sophistication, especially in analysis. When we started our collaboration with Dr. A. Gerstner [189], the aim of our relatively simple study was to explore the applicability of hyperspectral imaging for the analysis and classification of human mucosal surfaces in vivo/intrasurgically [189]. The larynx as a prototypical anatomically well-defined surgical test area was analyzed, as a gateway to other head and neck/upper aerodigestive tract imaging applications. Two main goals were addressed, (1) early detection of so far undiagnosed carcinoma and (2) intraoperative visualization of lesion borders.

There is no standardized scheme for the early detection of head and neck cancers worldwide. Although they often arise in anatomically accessible areas (using flexible endoscopes under local anesthesia), delayed diagnosis is common. Among many factors contributing to this there is a significant connection with the individual investigator’s experience: the eye—and the brain behind it—recognize best what it expects to see. Unfavorably, those patients at the highest risk (smokers and drinkers) have the lowest compliance with the examination. Therefore, it would help a lot to have some kind of marker that highlights cancerous mucosae. Those markers developed so far are either external dyes (rose bengal, toluidine blue) or fluorescence (autofluorescence or 5-ALA). None of these are either specific or sensitive enough for routine clinical use, let alone for screening.

Although recently some technologies have been developed for thorough “bloodless” investigation of upper respiratory tract mucosa (confocal laser endoscopy, optical coherence tomography, narrow-band imaging), a suspicious lesion has to be identified in an easy and reliable manner first. It would be completely unpractical to screen the entire mucosa with OCT or confocal endoscopy.

Hyperspectral imaging allows access to the underlying biological function of the mucosa by acquiring and analyzing its spectral signature. The biological function is expressed by the pattern of proteins, glycans, flavins, NADH, and other structural elements, which in turn shape the spectral emission. If this technology were more widely available for endoscopy (and faster—see below) it could be an invaluable adjunct or even eventually replacement to the naked eye for evaluating the upper respiratory tract. If it could be included in “microscopy” in surgical terms this technique could be used intraoperatively for resection of tumors: it could better delineate the border between the tumor and healthy mucosa. There were a number of hyperspectral endoscopes developed and reported on around the same time, but most of these were used in preclinical (animal) experiments.

What we found [189] in patients, by microlaryngoscopy with a polychromatic light source and a synchronous triggered monochromatic CCD-camera, in hyperspectral image stacks (from 5 benign and 7 malignant tumors) analyzed by established software (principal component analysis, hyperspectral classification, spectral profiles), was that the method had potential, i.e., that hyperspectral imaging can be applied to mucosal surfaces and their differentiation. In principal component analysis, images at 590–680 nm loaded most onto the first PC which typically contained 95% of the total information. Hyperspectral classification clustered the data highlighting altered mucosa. The spectral profiles (signatures) clearly differed between the different groups. This approach opened the way to analyze spectral characteristics of histologically different lesions in order to build up a spectral library and to allow non-touch, near-real-time optical biopsy.

In the decade since, this field has evolved very quickly. Improvements in endoscopic hardware integration with spectral options [190] were followed by automation of analysis software [191], and further advances in both allowing much better workflow and interpretation [192], made more impressive by adding deep learning to the arsenal [193,194]. The references in these studies show a vibrant, active field that could benefit from the new capabilities, beyond what is achievable with the competing narrow-band imaging [195], which technically speaking is subsumed and performance-wise surpassed by hyperspectral imaging endoscopy. Recent advances were reviewed in [196]. An additional flavor of hyperspectral endoscopy will be outlined below, based not on composite, phenomenological (but still nicely discriminatory) signatures, but rather on a well-understood physical feature of light impinging on tissue (Mie scattering [197]).

#### 5.3.2. Elastic Scattering Imaging Endoscopy

More than 85% of all cancers arise in the epithelium that lines the organs. Early lesions are almost impossible to detect, but such capability would greatly help lead to better outcomes. Cancers are complex and different from each other. One common denominator is that their nuclei are enlarged, crowded and hyperchromatic. A normal nucleus is about 2–9 µm in size, while cancer cell nuclei can reach 20 µm, with dysplasia in between. In tissue, nuclei are really the only major scatterers, and therefore a scattering measurement could give us information on the presence of cancer.

There is a very old theory of scattering in this regime (with the wavelength of light being about 1 to 1/10 of the scatterer’s size, and it was given by Gustav Mie in 1908 [197]. It is usually known as elastic scattering and states basically that the spectrum of light backscattered by particles (such as the nuclei) contains a component that oscillates characteristically with wavelength, and that this variation only depends on particle size (and refractive index). This knowledge/feature was neatly exploited by Backman et al. [198], who used a point-detection fiber endoscope and showed a very convincing ability to separate cancer and even precancer from normal tissue in the oral cavity, bladder, colon, and esophageal epithelia. Surprisingly, in spite of the relative ease and elegance of the method, and the fact that the scattering signal is intrinsic (i.e., requiring no preparation or contrast agent added), there appeared to be no follow-up on this work. Lack of clinical enthusiasm, in spite of the great promise shown, prompted us to conclude that endoscopic surgeons would prefer this information to be available in an imaging mode, where every pixel’s Mie scattering can be measured simultaneously, to guide intervention.

We developed such an approach: Elastic Scattering Spectral Imaging Endoscopy [199,200], based on a hyperspectral imaging endoscope that used all that we learned from our other spectral and multimode work reviewed here, calibrated and characterized it with model systems we prepared [201], and patented it [202]. The new endoscope (Figure 24) allows for the non-contact, rapid (sub-second) acquisition of polarized (both polarizations, to correct for the uninteresting specular reflectance from tissue) spectral images of tissue in vivo. The general intent was to enable exploration of a variety of optical contrast mechanisms (such as light absorption, reflectance, scattering, and fluorescence) in a search for new methods of early cancer detection in a clinical setting, but our first new implementation for cancer detection was based on the aforementioned body of spectroscopic work that employs elastic scattering (Mie) theory to estimate the size of bulk scatterers in a given medium—in our case, the epithelial tissue of lungs. Wavelength selection was by a computer-controlled fast monochromator (Polychrome). The rest of the system was built mostly from off-the-shelf commercially available parts, with the exception of the custom fiber, the synchronization electronics, and the AOTFs that later replaced the monochromator. The entire apparatus is packed onto a cart that can be safely wheeled around to various locations (such as a hospital operating room or an outpatient facility, see Figure 24). Custom software (EndoSpect) was written to gather the optical biopsy information for later analysis. This software operates in two distinct modes, Live and Biopsy. In live mode, a single optical frequency is used to estimate the biopsy area and range. A frequency of 620 nm was chosen because it was empirically determined to yield the best reflection from the lung tissue. Biopsy mode is triggered either by the computer operator, or by the surgeon using a foot pedal, and is

–Able to acquire the entire spectral range from 350–800nm (at 10 nm increments), in less than 0.5 s (8 ms per band). Spectral bands do not need to be in any particular order or be spaced a certain way, and indeed bands of interest can be repeated at different points during the scan.–Built for non-contact operation. The tip optics have a focal length of 10 mm, and a viewing area of about 8 mm in diameter at that range. No contact with the target is required or even desired. Indeed, the ability to image without direct specimen contact is especially useful since some tissues are easily traumatized.–Able to acquire of both parallel and perpendicular images simultaneously, allowing for these contrast mechanisms to be explored–Able to be used as a daughter endoscope in the instrument channel of an existing endoscope, to view lung or GI tissues in vivo–Able to be sterilized appropriately for use in a hospital setting–Using fast band-sequential spectral acquisition that limits intra-band interference and noise, as well as limiting movement artifacts; exposure times can be set so that the spectral image datacube has homogeneous signal-to-noise.–Able to be transported to the site of imaging/OR.

Two clinical trials at Pittsburgh hospitals were conducted with this instrument, looking for early lung cancer in at-risk populations. The system performed as designed, but no cancers were detected in a total of 62 patients. Although we believe the results to be correct and reproducible, we analyzed the areas where significant improvements could be made and found that, even according to theory (and extensive modeling we conducted), a much better spectral resolution would be very helpful.

With support from the US Air Force, and with collaborators [203] we developed an ultra-spectral light source consisting of a supercontinuum laser (Leukos, Limoges, France) and liquid crystal arrayed microcavities we designed for ultra-narrow wavelength filtering (FWHM~0.1 nm). These use a picoliter volume Fabry-Perot-type optical cavity filled with liquid crystal that, upon application of an electric field, achieves this delicate spectral tuning with high finesse. With this sub-nanometer spectral resolution, we measured spectral oscillations from scattering particles to validate simulation results. We also measured chicken liver and breast muscle and pig skin and muscle, to compare with results from the literature. The main application was nuclear size estimation in biological tissues, for non-invasive assessment (optical biopsy) equivalent to careful histopathology (see Figure 24, right) [204].

#### 5.3.3. Multimode Imaging System for Minimally Invasive Surgery

With support from the US Navy, we undertook designing and building a new, very versatile, and complete multimode endoscopy system for use in some experimental procedures in an operating room that relies on optical biopsy. We intended to give it all of the features and advantages that can be derived from the work previously described here (see above). We ended up (see Figure 25) with 11 modes, fast switchable and computer-controlled. Coordination with OR procedures was necessary and completed, and the system performed well but was unwieldy because of incomplete integration. This motivated us to develop a new design and implementation for a digitally controlled OR of the future (see below) that can more easily accommodate such new instrumentation.

### 5.4. Theranostics

#### 5.4.1. Intraoperative Neurophotonic Detection and Guidance

Advances in image-guided therapy enable physicians to obtain real-time information on e.g., neurological disorders such as brain tumors to improve resection accuracy. Image guidance data include the location, size, shape, type, and extent of tumors. Recent technological advances in neurophotonic engineering have enabled the development of techniques for minimally invasive neurosurgery. Incorporation of these methods in intraoperative imaging decreases surgical procedure time and allows neurosurgeons to find remaining or hidden tumors or epileptic lesions. This facilitates more complete resection and improved topology information for postsurgical therapy. We reviewed the clinical application of recent advances in neurophotonic technologies, highlighting the importance of these technologies in live intraoperative tissue mapping during neurosurgery [205]. While these technologies need further validation in larger clinical trials, they show remarkable promise in their ability to help surgeons better visualize the areas of abnormality and enable safe and successful removal of malignancies. Nearly 700,000 people in the United States are living with a primary brain tumor, and ~80,000 more are diagnosed every year. Glioblastoma multiforme (GBM) is the most common infiltrating primary brain tumor in adults and one of the most aggressive cancers with severe negative effects on patients and the health system. Many other brain tumors are noninfiltrating and can be removed without damaging normal brain tissue. GBMs represent 16% of all primary brain tumors, and despite intense multipronged therapy consisting of surgery, radiation, and chemotherapy, the outcome of patients with GBM remains very poor, due to the blood–brain barrier that blocks therapeutic agents from the tumor site and the chemo-resistant and radiation-resistant nature of glioma-initiating cells. Tumor delineation is very important during surgery, and it is an extremely difficult and subjective process, requiring a skilled neurosurgeon to interpret sometimes inaccurate preoperative imaging to achieve near-complete resection. Because of post-imaging changes that occur during procedures, there is an ongoing need for real-time, intraoperative detection.

When considering the resection of a tumor, one must be aware of the plastic nature of the human brain. Brain tissue is extremely flexible and deforms following dural opening. This phenomenon, known as “brain shift,” is affected by many factors such as mechanical tumor resection, evacuation of cystic components, gravity, pharmacologic responses, osmotic shifts, drainage of cerebrospinal fluid, and other surgical effects. Brain shift leads to significant challenges in the use of conventional neuronavigational systems, which use preoperatively acquired images and stereotaxy to depict the anatomical location and estimate the three-dimensional (3-D) extent of brain tumors. Many optical technologies are also beginning to be applied in this context, including infrared (IR) thermal imaging, Raman spectroscopy, fluorescence spectroscopy, and imaging, fluorescence lifetime spectroscopy and imaging, and OCT. We focused on clinical implementations of these optical technologies, as intraoperative guidance tools in human neurosurgery, with emphasis on their respective merits and challenges, and on the achieved real-time classification, detection, and guidance.

The field of brain mapping has moved far from just being defined as brain imaging into a field that now includes functional mapping, brain connectomics, cellular/molecular, and metabolomics imaging. This is due to recent advancements in the field of optics and imaging in general, which include molecular and metabolomic imaging and spectroscopy. Biophotonics is being applied more and more to resolve neurological problems, creating the burgeoning new field of neurophotonics. Today the entire field of intraoperative brain mapping is moving toward real-time, biomarker-based, meta-data analytics-based assessment of tissues because the field of neurosurgery, in general, is moving away from the use of major devices and invasive approaches to personalized nanoscale diagnostics and therapeutics. Therefore, such advancements in the field of drug discovery/nano-neuroscience, neurophotonics, genomics, and metabolomics are providing enormous opportunities for the introduction of intraoperative tools that are extremely high-resolution, capable of visualizing cell(s), in real-time (up to a few nanoseconds), and biomarker-based. This has revolutionized the field of intraoperative microscopy and enabled neurosurgeons and physicians to better map the human brain in real-time.

Of the emerging methods, due to the complex and dynamic tissue environment, fluorescence lifetime imaging spectroscopy showed particular promise, as it is practically independent of signal intensities. Moreover, Raman, fluorescence, and absorption spectroscopy are sensitive to different aspects of molecular and atomic structure and can provide complementary information about tissue characteristics. Multimode systems that combine these techniques may provide even greater sensitivity and specificity for brain tumor demarcation and intraoperative surgical guidance. Every technology has its own limitations but together they have great potential to unlock the mysteries and specifics of human brain function and disorders. We believe [205] that integration of neurophotonics with more traditional imaging such as ultrasound, CT, MRI, fMRI as well as with nanotechnology, cellular therapeutics, supercomputing, artificial intelligence, and other cutting-edge advances will revolutionize the way neurosurgical cases are managed in the future.

#### 5.4.2. Surgical Theranostics: Coupling Detection and Intervention in Time and Space

As we prepared to bring the best of our biomedical optical technologies to the operating room, we tried to consider all strategies, including new ones. We knew that ideally, any powerful and surgeon-adoptable imaging method should (a) be minimally invasive or, preferably, non-invasive; (b) have high spatiotemporal resolution; (c) be highly discriminating; (d) use no contrast agents; and, ideally, (e) be able to couple detection/diagnosis to any intervention, as tightly as feasible in space and time. It became clear that the methods discussed previously, confocal (for (a) and (b)), spectral (c), and autofluorescence [206,207] imaging (d) are good candidates, but additionally, a new possibility subsuming these arose, to address (e) above: scanning multispectral confocal imaging.

Working with Omega Optical [208,209,210,211], we developed a spectral confocal imaging workstation based on a technology originally designed for telecom (wavelength division multiplexing in the near-infrared). The setup is shown schematically in Figure 26 (left). Lasers were pulsed, at 405 or 488 nm, with about 2 mW at the sample, a 0.4 µm spot size, scanned at 10 frames/second over a field of view adjustable 10–108 µm, with a working distance of 80 µm, and an NA of 0.8.

The heart of the system, our fiber optic spectrometer is based on a serial array of reflecting spectral elements (labeled λ_1_, λ_2_, λ_N_ in Figure 26), delay lines between these elements (fiber loops in Figure 26), and a single photomultiplier tube. After excitation by a laser pulse, broadband fluorescence from a sample propagates into the array via the confocal aperture. Light of the shortest wavelength band (λ1) reflects from the first element, and light of the Nth wavelength band reflects from the Nth element. Each wavelength is mapped into a specific time slot. The two-way propagation time in each delay line is equal to the length of the laser pulse (about 166 ns for a 15-bin system). A plot of the detector’s signal versus time contains the laser in the first bin, and then fluorescence of various distinct colors in the subsequent bins. The spectral elements can be fabricated in two ways—fiber tips can be coated with interference filters to create >10-nm wide spectral slices, whereas fiber Bragg gratings can be written into the fiber core to create 1- to 2-nm-wide spectral slices [208,209]. Most spectrometers employ one grating that disperses light spatially across N detectors or pixels. Our approach employs N tips or gratings that distribute the light temporally against one detector.

The key advantages of this design are (a) speed—a spectrum is acquired during the dwell time of each pixel in the spatial scan; (b) wavelength bin centers and band-pass widths can be arranged with varying widths and spacings matched to a given application. The spectral widths and spacings in designs based on single bulk gratings are constrained by the diffraction equation; (c) the wavelength separation method has no influence on spatial scanning fidelity. A map of a given color will spatially register with a map of another color; (d) each polarization state is reflected in a similar manner; (f) the new design is compatible with confocal optics.

For spectral analysis and display, one useful colorization formula is based on the spectral angle mapping (SAM) algorithm that computes the cosine of the angle between the signal (S) and reference (R) vectors by computing the dot product of the two vectors and dividing by the product of their absolute magnitudes. Our software and firmware can determine the SAM angle for each pixel in a confocal scan and colorize the resulting image according to the magnitude of the angle—all in real-time. When the cosine is unity, the S and R vectors are perfectly matched. A key advantage of SAM images is that variations in brightness (due to fluorescence intensity or the degree of focus) do not impact the image. This is because SAM images are based on the angle between the two vectors as opposed to the magnitude of the vectors. Useful SAM images require appropriate reference vectors. For example, we developed reference vectors for three quantum dots that emit in blue, green, and red.

The multispectral imaging system has been designed, built, and tested for the spectral mapping of intrinsic tissue fluorescence (autofluorescence). The spatial resolution is sufficient for cellular-level imaging and has been used so subsequently [210]. The spectral resolution is suitable for detecting the spectral signatures of cancer. The temporal resolution is sufficient for avoiding the effects of cardiovascular and respiratory rhythms of living subjects. The approach is similar to confocal microscopes that use a rotating filter wheel, except that the filters are effectively rotated at the speed of light. We used the system on several samples, including fluorescent beads, quantum dots, fixed tissue, ex-vivo tissue, and in vivo tissue. SAM analysis has revealed healthy versus diseased tissue and can be implemented in real-time. Our approach is also a cost-effective design with a resolution appropriate for detecting intrinsic fluorescence in a clinical setting. Armed with suitable spectral libraries, our ultimate goal is to enable preventive healthcare and the early detection of disease via optical biopsy. The technique would also allow surgeons to assess surgical margins in real-time, by sensing the presence of cancerous cells sur- rounding an obvious tumor. For example, our multispectral imaging endoscope could be used in the intraoperative assessment of lumpectomy margins, and a representative case of such theranostics (on a rat model of breast cancer) is illustrated in Figure 26 (middle and right).

Mapping spectral bands into the time domain can be applicable to microscopy, endoscopy, and cytometry. One can imagine other applications involving fluorescence, two-photon spectroscopy, and Raman spectroscopy with a future impact on cytomics, histomics, and clinical settings [208,209,212].

### 5.5. Sensitivity and Specificity

As mentioned, our guiding principle for evaluating the usefulness of a biomedical optics method is its applicability to a specific, yet largely unmet challenge that is clinical in nature. We have chosen our fields of experimentation accordingly, and the work reviewed above illustrates the technologies needed, and the (new) level of performance achievable with advanced biomedical photonic tools.

One of the most accepted and thus widely used ways that clinical practitioners assess the success of an approach is by sensitivity and specificity. There are learned articles about the definitions and fine points of these, but in our simple terms, sensitivity means “(to what extent) can you find a problem if it is there”, while specificity means “(to what extent) can you tell a problem from a non-problem”. Obviously, one requires simultaneously high sensitivity and specificity for best results (100% sensitivity is reached if one decrees everything is a problem, but that pushes specificity close to 0%).

Table 2 below summarizes our results from the body of this review. It shows the desired simultaneously high sensitivity and specificity, for both preclinical and clinical applications, usually further improved by more sophisticated analysis of the same data. We believe that the main points it illustrates are that (a) spectral/hyperspectral imaging is a powerful method with great diagnostic performance and promise and (b) multimode imaging has even higher performance (often 100%/100%), provided that the right modes (and the number of modes) are combined in the right way, matching the difficulty level of the problem addressed.

### 5.6. The Operating Room of the Future

In order to have an impact, our (and others’) new devices need to be functional in the ultimate clinical environment, the Operating Room (OR). Our overarching goal of improving the way surgery is performed by strategically bringing relevant biomedical optics-based technological advances into the OR. For this we had to address (A) the need for a new type of information management architecture, and (B) the need to start with a simple implementation, focusing on the user interface. Connecting these elements makes it possible to “grow” an OR that combines versatility, lower cost, and other desirable features, and is ready to meet the users’ needs as they evolve. The need for (A) is quite obvious if one examines the current state-of-the-art (see Figure 27, left). The OR is proprietary, vendor-specific, and heterogeneous, which makes it difficult to add to, and (frankly) messy, which does not help with workflow. The OR of the Future will have a multitude of information sources, including vital sign monitors, endoscopes, cameras, ultrasonic imagers, and X-ray and other imaging equipment. In addition, it should provide easy access to pre-operative data, including MRI and CAT scans, medical histories, data from previous procedures, pathology results, etc. Furthermore, it should be possible to monitor the procedure remotely and to consult with external specialists. Currently, this information exchange is hampered by a multitude of incompatible interfaces and by the lack of an integrated, modular, extensible information management system. We researched the structure of the current system, identified its deficiencies, and proposed a simpler, cheaper, yet more powerful architecture that largely uses open, standards-based, commercial components instead of dedicated, proprietary, closed, special-purpose units. The OR of the future will include a multitude of medical devices, that provide information to the surgeon and that need to be monitored and controlled. This is both a quantitative and a qualitative change from the current OR environment.

Currently, the number of medical devices, such as vital sign monitors, endoscopes, ultrasound imagers, X-ray equipment is relatively small, which means that it is reasonable that each device is designed to work stand-alone, with various interfaces added to it as secondary options. Each device has its own set of controls, its own style of interaction, and its own set of formats for storing and/or exporting data, and thus it is not surprising that they do not operate well together. However, for optimal activities of low bandwidth devices (vital sign monitors, ventilators, insuflators, electrical or ultrasonic scalpels, infusion pumps, sensors for pressure, temperature, flow, etc.) we need universality, extensibility, reliability, availability, serviceability, ruggedness, and even some redundancy. The options are RS232/422/485, GPIB, P1394/Firewire, USB, Ethernet, WiFi, and LAN technologies. For high bandwidth devices such as video cameras and other image-producing devices (as described above), analog (NTSC video, S-video, RGB video, digital TV formats, P1394) was the norm for a long time. Despite the large number of incompatible outputs from these imaging devices, they tend to have one characteristic in common: they expect a monitor at the other end of the wire. Due to the limitations of past technologies, image data tends to be treated as video streams that originate from an image source (camera) that then continuously flows to an image sink (a display or recording device) over a dedicated medium. There is no feedback from the sink to the source and control and auxiliary data is transferred by other means (out of band). This led to the emergence of crossbar switches that can connect any image source to any image sink. With some borrowed technology from TV studio gear, it is possible to scale the image streams, add overlays and have some very limited composition capabilities.

Nowadays, most image data in the medical domain tends to be generated digitally. MRI, CAT, and Ultrasound machines form images computationally. Cameras may appear like analog devices but tend to digitize the raw sensor signal either directly on the sensor chip or on a nearby support chip. Digital signal processing takes care of sensor imperfections, adjusts gamma, corrects the white balance, adjusts exposure and gain, performs various filter operations, etc. At the end, this data can be converted back to an analog signal. Practically all current and future devices will have the data stream in digital form internally. There is another aspect of this design that is in conflict with current technology trends: display devices are also becoming more digital in nature. For example, a flat panel display may feature 1200 × 1600 pixels or more that are refreshed 100 times per second. In the current video-stream-centric architecture, this requires an enormous amount of bandwidth that needs to be sent to the screen, even if the screen displays a static image or a graphical representation of a vital sign monitor with perhaps an ordinary TV picture.

We proposed and implemented the use of Ethernet LAN technology for all devices in the OR. This treats all data, low and high bandwidth streams in exactly the same fashion. Each device is connected to a COST TCP/IP router via a dedicated cable that runs at the appropriate speed. At this point in time, a single CAT-6 cable running a 1 Gbit/sec is sufficient to handle each device. However, there is nothing in this architecture that is tied to a particular speed. 1000bT Ethernet routers routinely adapt to 10bT and 100bT devices and TCP/IP routers tend to be flexible and modular so that a device may either use multiple connections concurrently or use 10Gb/s links without having to change the rest of the system. Having a single type of cable that does it all would greatly simplify the OR cabling. Years ago, the FLIR thermal camera was an early example of this approach, by relying on a single gigabit Ethernet connector. All functions of the camera are controlled by this one wire that also transmits the thermal image at up to 500 frames/sec. There is no need to shoehorn the data is some sort of NTSC video stream that loses the dynamic range and has the wrong frame rate and aspect ratio.

Any well-conceived design/implementation should yield some practical advantages. The better flow of information and resources and enhanced versatility should help the surgical teams greatly. Technology developers should be helped in bringing new devices more seamlessly into the OR, and this includes experimental equipment such as those featured in this review. Benefits to the manufacturers include up- and down-gradability of the product line, self- and remote diagnostics of the system (significantly lowering support costs), hot-swap stand-in control computers that can take over from a failing one in a fraction of a second, and the ability to have all customers on the same version of the same control software running the OR equipment (this could be simultaneously delivered to all customers via networking, upon successful completion of a new version by the company).

With support from the US Navy, we built a Digital OR of the Future, and added some of our advanced endoscopes (see Section 5.3 and Section 5.4 above), as well as some other features that we believed a modern OR should have: a non-contact surgeon-computer interface using hand gesture recognition [213], and a digital surgical checklist (integrated with the minimally invasive surgery workflow), which appealed to us since it was shown to drastically improve patient outcomes in all clinical settings tried [214].

Finally, there is a clear and intensifying trend to bring robotics (such as Intuitive Surgical’s DaVinci robot) into the OR [215]. Best adoption scenarios should be based on a flexible, versatile digital OR of the kind we advocate and implemented.

## 6. Discussion


*“The future is here. It is just not evenly distributed.”*
William Gibson

The science/technology/clinical domain reviewed here is, without a doubt, evolving extremely quickly and one needs to keep track of new developments to stay relevant. Looking back every 20 years [216] would just not do—we have to make sure that we constantly review, learn, and strategize.

This review is, like most, somewhat subjective, based on its main intention: outline, on the background of evolving biomedical optics, how we arrived at and implemented the multimode imaging concept, how we picked its most translationally relevant applications, and how all this affected the ability to bring the new approach to the clinic.

### 6.1. Optical Bioimaging Adoption in Medicine and Surgery

The adoption of optical bioimaging in medicine has been slow for several reasons. First, imaging structures deep within the body presents a significant challenge for visualization by optical methods. In the body, which is highly opaque and scatters light, one cannot use UV light for excitation, or high laser intensities, or any other extreme conditions because of poor penetration and potential damage from the molecular all the way up to the gross tissue level. There is also a marked paucity of proper contrast agents, particularly for in vivo intrasurgical applications. Most of the agents currently used were introduced five decades ago and fall short of today’s standards in both safety and efficiency, not having been developed with current technological capabilities in mind or optimized for machine vision. The cost of properly testing a new agent is prohibitive, as regulatory hoops require their safety and efficacy verification in a very broad set of circumstances, to newly raised standards. However, one cannot deny the importance of labeling biomarkers for progress, as biological objects lack intrinsic contrast. To quote Floyd Bloom, a former editor of *Science*, *“The*
*gain*
*in*
*brain*
*lies*
*mainly*
*in*
*the*
*stain”.* This is why we prefer, believe in, and have concentrated on what can be achieved using intrinsic signatures (ideally molecular, but phenomenological can also help—see Section 3 above).

First, at least six optical methods are potentially available for discriminating normal tissue from abnormal with high spatiotemporal imaging resolution, without using contrast agents: spectral reflectance, autofluorescence, fluorescence lifetime, Raman, Stokes-shift, and elastic (Mie) scattering imaging. Each method has potential advantages and disadvantages as a stand-alone technique, and several such methods have already been proven to be useful adjuncts to surgery. The effort to test each of these individual methods against each other would be quite significant and result in delays in the development of clinically meaningful tools. However, the capacity to combine these methods in a single platform and subsequently analyze the potentially complementary and even synergistic information gained by multimode imaging presents a unique opportunity for the advancement of this field. The use of a relatively recently developed and commercially available supercontinuum laser light source to enable and deploy all such imaging methods in a single platform may prove important in advancing this agenda. The result should be a hyper-reality (interpreted) image available to the surgeon—in addition to the usual images, endoscopic, open-field, or PACS—in order to assist with decision-making, by providing the equivalent of advanced intrasurgical pathology, in real-time (the by-now famous optical biopsy). Once the size, shape, orientation, and location of the lesion of interest (tumor, lymph node, etc.) are assessed by digital imaging, it becomes easy to “paint a target” for intervention such as robotic excision or laser ablation.

Second, there is a need for regulatory oversight and approval in point-of-care instruments. Laser eye surgery achieved fast success because of the quality of the instruments, the optical nature of the target application (ophthalmology), and the relatively minor modifications needed for adoption. In gastrointestinal (specifically, small bowel) imaging, untethered optical capsule endoscopy is an unexpected, inventive, and quantifiable technology [217] that has been approved by the FDA in record time because of its clear advantages over established procedures.

Third, the medical community has traditionally been much more conservative than the research community in adopting new technology; physicians still largely rely on their own senses and intuition in diagnosis and treatment. Egregious examples include clinical pathology, where the entire surgical procedure and its timeline (including subsequent additional surgery because of false positives that can run as high as 30%) is set by the subjective call of a pathologist, and melanoma diagnosis, where the typical presurgical examination is about as advanced technologically as it was in antiquity—looking with the naked eye. Quantitatively speaking, some of the most important applications of optical bioimaging (e.g., as a tool in pathology) are still carried out as they were a century ago, on fixed specimens with stains that have not been standardized. In an era when we trust satellite imaging to predict global meteorology, optical fibers to beam communications around the world, and lasers to manufacture goods in factories and guide missiles with ‘reduced collateral damage’ (to say nothing of what smartphones are capable of) we somehow cannot bring ourselves to pervasively use advanced optical imaging to diagnose and pinpoint disease where and when it matters, in the OR.

### 6.2. Translational Biophotonics: Moving into the Operating Room

We believe that the development of photonic tools and methods for medicine and surgery should emphasize non-destructive/non-invasive uses, in vivo. We were recently able to identify and address two of the main limitations that, in our estimate, have prevented the penetration of advanced optical imaging technologies into the operating room. The first obstacle to translational research bringing optical imaging from cells on a slide or even preclinically in vivo, to use on a patient in an operating room is the near-impossibility of introducing new optical contrast agents into human clinical use. Older, accepted probes are unwieldy, non-specific, or both, while new, custom-designed ones—very successful in animal work—are potentially toxic and thus unusable due to real, and certain regulatory concerns. We have therefore concentrated on those optical imaging approaches that do not require contrast agents; this necessitated advanced methods, using novel photonic technologies (spectral [218], lifetime, non-linear and coherence-based approaches, applied to light emanating from the specimens: reflected, scattered, transmitted, and emitted as fluorescence and luminescence). The synergetic use of these imaging modes promises to be the most effective for detection, diagnosis, and surgical guidance. Our overall approach to intrasurgical optical imaging is thus based on a multimode implementation and is endoscopically optimized for improved minimally invasive surgery with earlier detection of abnormalities. This yielded elements of a new armamentarium for surgery.

For addressing the second obstacle: With imaging fulfilling its dual role of better describing both anatomy and physiology, in vivo cytometry/intrasurgical histopathology-equivalent molecular and cellular imaging is achievable, as is a closer spatio-temporal connection between imaging (for detection followed by diagnosis) and intervention. This should also allow the optimization and computer-assisted decision-making and guiding of surgical activities, but additionally requires an ability to seamlessly introduce the new technologies into a clinical environment of the future. We completed a new design for such a venue, centered on an all-digital operating room optimized for high-tech flexibility and ready for the deployment of new and advanced laser-based methods into clinical practice (thus addressing the second limitation of translational research).

Our recent results are representative of both translational challenges and their technical solutions, and of some major application areas, including early cancer detection by spectral reflectance and autofluorescence; progression quantitation by improved optical coherence-based imaging; brain cancer and cancer stem cells; nano(photo)medicine approaches to cancer detection and treatment; the imaging of very fast calcium transients in a neuromuscular junction; hyper-spectral Mie (elastic) scattering imaging for endoscopic guidance; and the design and use of an advanced multimode imaging endoscope, partially based on spectral imaging and full-field multiphoton imaging. Some regenerative medicine applications were also successful, including monitoring of stem cell fate *in vivo*, and cartilage, heart, Alzheimer’s Disease, and image-guided surgery work. The processing and analysis of spectral and multimode images is always challenging, and our advances in this area (with emphasis on contextual and speedy visualization), enabled an application to, e.g., the in vivo delineation of Hirschsprung’s disease for surgical intervention, potentially saving newborns’ lives.

### 6.3. Avoiding Errors

Medical errors are a significant cause of death in the US, no matter whose widely divergent numbers are used, equivalent to at least a couple of large planes falling out of the sky every day. Other areas of high-risk human activity (such as flying planes full of people) have learned how to deal with the need to reduce such risk. A very realistic theoretician of risk mitigation, Dr. Reason [219] is basically stating (see Figure 28) that (a) we need to interpose something between us and the hazard; (b) because we are human, everything we do is limited/flawed, and thus the barrier we interposed looks like a slice of Swiss cheese (with holes) and, most importantly (c) the simplest way to deal with this is to stack the “cheese slices”, because the holes in them do not align, and this approach should significantly reduce the risk of the hazard reaching us through the stack (Figure 28 left). We agree with this concept, and believe that the biomedical optical imaging equivalent is multimode imaging, but propose that our methods/modes should not just be overlapped, but rather funneled, in the sense that the sequence needs to start with lower resolution, faster ones, and—based on what they show—zero in on areas of concern with more and more specialized modes, so that, e.g., we do not need to multiphoton imaging in the whole body, only in areas pinpointed by several other methods working as a relay (see a schematic in Figure 28).

### 6.4. Biophotonics in Pandemic Times

We are currently still in the middle of a pandemic that resulted in painfully great loss of life, severely impacted the global economy, and re-arranged everyone’s priorities. A few observations to connect these happenings to the subject matter covered in this review are as follows:

While the world appeared unprepared for COVID-19, the realization of the dangers posed accelerated action. Dealing with the SARS-CoV-2 virus and its variants was prioritized over all else clinical, and both new diagnoses and treatments were developed and, after regulatory approval, were applied to humans in record time. Thus, when critically needed, translational scenarios—from molecules to humans—can happen very fast [220].

While clearly disciplines like virology, immunology, epidemiology, and public health are needed, relied upon, and deployed in such instances, the two biggest technological components in these efforts were molecular sciences (broadly defined) and optics. The speed with which the sequence of the virus, in vitro diagnostic testing (molecular, antibody, antigen) of patient specimens, and vaccine development came online was, at least historically, impressive. Optical methods, most originally developed for other purposes, play a vital role, from sequencing to point-of-care and beyond. Accuracy, speed, ease-of-use have all been addressed, but the great remaining challenge remains the broad worldwide deployability of diagnostics and countermeasures [220]. It is likely that, particularly for underdeveloped areas, ubiquitous smartphones will constitute part of the solution. Their optics and electronics might constitute the piece of infrastructure badly needed in this context.

Some of the new treatments proposed are somewhat unexpected and can potentially address the dreaded virus variants. Potent neutralizing nanobodies have been tested and found effective, using structural tools such as advanced cryoelectron microscopy [221]. It would certainly be desirable to image a virus in action, in living cells with omnidirectional nanometer resolution, with the method described in Section 2.2 above, and in living animals, as in Section 3.2, for investigating mechanisms.

For the much-needed differential diagnostics (for the current pandemic and future challenges), it would be highly desirable to test for multiple diseases/pathogens/conditions simultaneously. Optical multiplexing [222] appears to hold the most promise in achieving this.

There are more lessons to be learned here so that we are better prepared for the next crisis. Let us just state that advanced technologies are important and, if well-understood and implemented, they can be re-purposed fast, affordably, and effectively to address most challenges [223]. Doing so early and strategically rather than under pressure is preferable, for eliciting faster, better outcomes.

## 7. Conclusions


*“The best way to predict the future is to create it.”*
Dennis Gabor (Nobel laureate)

We are all using technology extensively, and most of us are in awe of some of its more impressive achievements. Wouldn’t it be wonderful if all life-saving activities on Earth had the sophistication of a Mars rover? This, of course, will not happen by itself, or soon, but we have to work towards it. Some of the tools discussed here (such as lasers, LEDs, digital cameras, computer networks, spectral imaging, and so on) were developed by people at NASA, or DoD, or telecom: gifted, focused, dedicated, and with a lot of resources. This made the results more available and affordable. The incredible recent developments in smartphones and other mobile technologies and capabilities are in the same direction, in a quantitatively more massive and approachable way. What is needed now is continued dedication, coupled with a certain clarity of purpose, to bring these types of tools to bear, in a strategic way, on the betterment of the human condition, and specifically towards fighting disease. We might not even be able to visualize how an operating room should look in 20 years, but we should note that the aforementioned Mars rover Perseverance has *25* cameras, in addition to its lasers, drill, and mini-copter, so that we can image dust millions of miles away far better (in all regards) than we can image inside a human in serious health trouble in our best hospital. To those of us who believe in seeing (better), maybe we took a wrong turn somewhere, or, more likely, the right people have not been talking to the other right people—we have to ensure they do. As to how things might look in the future, here are some words of wisdom from one of the visionaries of nanotechnology, Christine Peterson: *“If you’re looking ahead long-term, and what you see looks like science fiction, it might be wrong. But if it doesn’t look like science fiction, it’s definitely wrong.”*

## Figures and Tables

**Figure 1 molecules-26-06651-f001:**
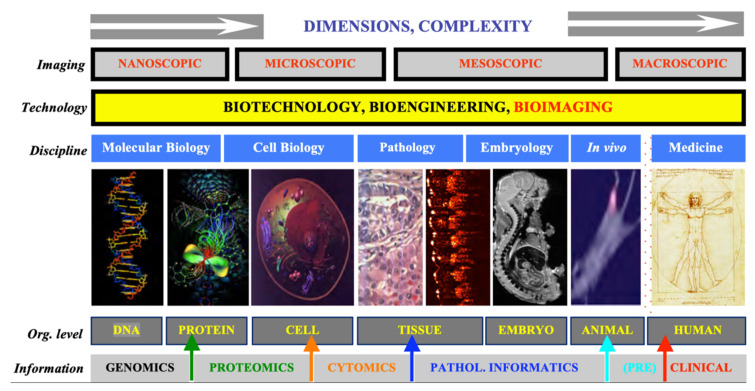
Translational research from molecules to humans, highlighting the central role of optical bioimaging. Pharma/biotech are focusing on DNA/protein-level solutions (green arrow), our best imaging is at the cellular level (orange arrow), but most advanced optical imaging stops at the preclinical level (turquoise arrow). Diagnosis and clinical decision-making are based on pathology (blue arrow), while clearly the need and resources peak at the clinical level (red arrow). We aim to superimpose these arrows towards the clinical end, overcoming the translational brick wall (dotted lines to the right). *We focused on advanced, multimode methods that help overlapping the left arrows onto the right two (rather than the reverse, as in biopsies), by expanding quantitative molecular imaging from (sub)cellular microscopy to preclinical studies and the clinic.* (modified from [4]).

**Figure 2 molecules-26-06651-f002:**
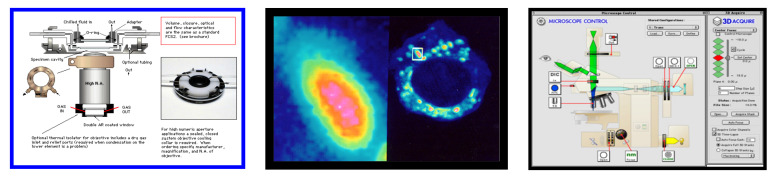
Quantitative imaging of plasma membrane and mitochondrial potentials in living cells. Left panel shows the live cell chamber we developed [10] to maintain the cells properly perfused, at strictly controlled temperatures, and allow fast addition of chemicals. The middle panel shows cellular (right) and (selected) mitochondrial (left) images, following Nernstian staining with TMRE. White box corresponding to center left image is 2 µm × 2.5 µm. Right panel highlights the 3D acquisition software and user interface [11].

**Figure 3 molecules-26-06651-f003:**
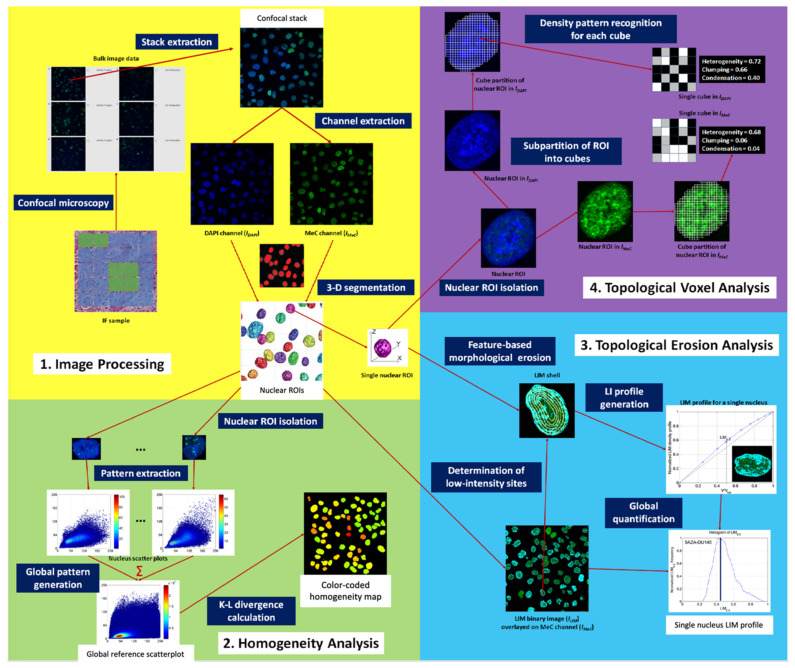
Graphic outline of main steps for 3D-qDMI image analysis. The first step of image analysis is the delineation of nuclear ROIs through 3D imaging. This involves the extraction of stacks from the confocal image, a seeded watershed method to create clearly defined 3D shells, and three different analytical modules: First, an assessment of homogeneity in all cell populations: through pattern extraction and creation of scatter plots for individual nucleus and a reference plot for the entire population, Kullback-Leibler (KL) divergence is used to categorize every cell based on its similarity to the population. The associated color-coded map allows easy recognition of outlier cells. Second, the topology of a nucleus is analyzed by peeling the nucleus into numerous shells. The shell is peeled from the periphery to the center by anisotropic shells defined by nuclear sizes. Within each shell, the concentration and distribution of low-intensity pixels are identified and tabulated to delineate the nuclear topology. Third, the condensation level of the genome is analyzed by dividing a nuclear ROI is into cubes of constant voxel sizes. The distribution and location of low- and high-intensity pixels are analyzed for each voxel. These results are later combined for further analyses by comparing the condensation patterns for DAPI with associated changes in methylation. Please see ref. [21] for more details.

**Figure 4 molecules-26-06651-f004:**
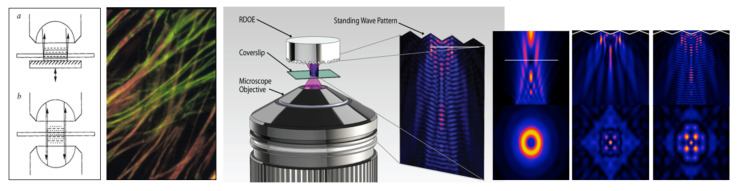
Standing wave (left) and Omnidirectional standing wave microscopy (OSWM) (right). Left: standing wave (SW) (**a**) and 4π (**b**) optical excitation schematics, and pseudocolor SW images of fluorescent actin fibers in a cell Red/green), spaced axially 35 nm apart, and easily resolved. On the right, the structured standing wave pattern resulting from interference of the incident illumination and light reflected from the RDOE excites fluorescence at the specimen plane. An expanded view of the structured standing wave pattern is shown at the right of the objective. In the rightmost insert images, the same patterns are shown (L to R) without the RDOE; with the RDOE, and with the RDOE, after interference. The delicate interference patterns can be gently moved (e.g., piezoelectrically) to selectively illuminate very small specimen voxels.

**Figure 5 molecules-26-06651-f005:**
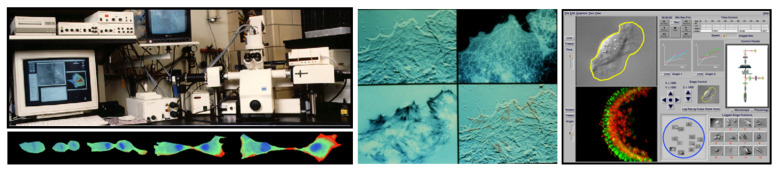
Multimode optical microscopy. (**Left**): First multimode microscopy workstation, with 4 camera ports, and custom image acquisition and processing software written from scratch for robotic-level automation of multimode time lapses. (**Middle**): Typical 3T3 cells imaging dataset, with (clockwise from lower left) reflection interference dynamically showing adhesion points of live cells, differential interference contrast for fine details of leading edges, fluorescence of selected intracellular component (endoplasmic reticulum) and an overlay of the first two panels. (**Right**): Intuitive graphic user interface for setting up automated acquisition in four dimensions and simultaneous monitoring of up to 24 selected locations by stage automation in 3D. After completion, the workstation ran 24/7 for years.

**Figure 6 molecules-26-06651-f006:**
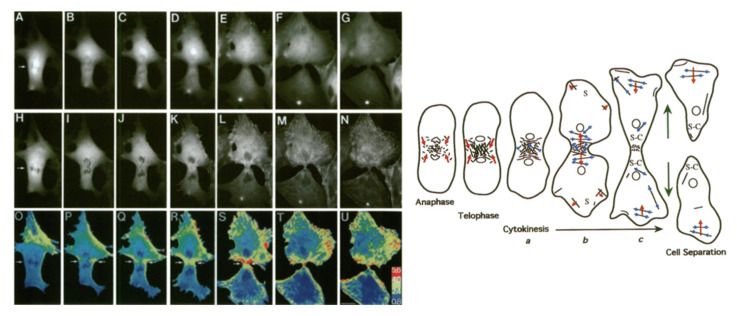
Myosin II dynamics in cytokinesis. (**A**)–(**G**), (**H**)–(**N**) and (**O**)–(**U**) represent the same time sequence for a volume fluorescence probe (top), Myosin fluorescence (middle) and Myosin II concentrations (in pseudocolor, bottom). They show a dramatic increase at the cleavage furrow, supporting a cortical flow/solation-contraction coupling mechanism [51].

**Figure 7 molecules-26-06651-f007:**
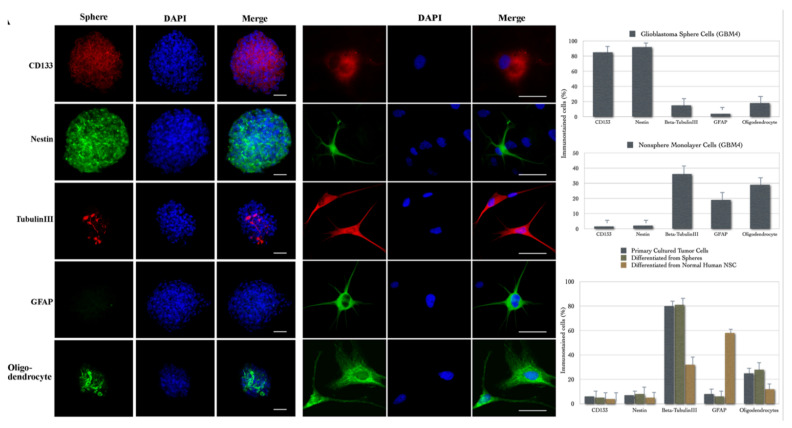
Glioblastoma cancer stem cells discovery and characterization by imaging. (**Left**): Single-mother-cell-derived glioblastoma spheres express NSC markers as well as lineage markers, but with different staining profiles from non-sphere tumor cells. Glioblastoma spheres derived from a single mother cell were stained by NSC markers CD133 (red) and Nestin (green). Lineage markers for neuron, b-tubulin III (red) and oligodendrocyte, myelin/oligodendrocyte (green) were stained, but staining for the astrocyte marker, GFAP, was not observed in the glioblastoma sphere. Cells were located by counterstaining with DAPI (blue); (**Middle**): Glioblastoma sphere-differentiated progenies show a phenotype similar to the parental tumor, but different from that of normal human neurospheres. Differentiated progenies of glioblastoma spheres were stained for the NSC markers, CD133 (red) and Nestin (green). The cells were also stained for lineage-specific markers of neuron (red), astrocyte (green), and oligodendrocyte (green). (**Right**): Stem-like cells derived from glioblastoma express neural stem cell lineage markers (top) and are multipotent (bottom). Scale bars = 50 µm throughout. Modified from [55], where more details can be found.

**Figure 8 molecules-26-06651-f008:**
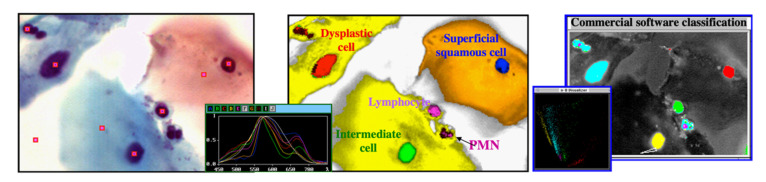
Hyperspectral imaging cytopathology. (**Left**): RGB image of typical Pap smear. This specimen was spectrally imaged and classified using our analysis software (**middle**), revealing the cell types present. The same segmentation and conclusions were obtained by a commercial spectral analysis package (ENVI) using a clustering algorithm (note the different pseudocolors).

**Figure 9 molecules-26-06651-f009:**
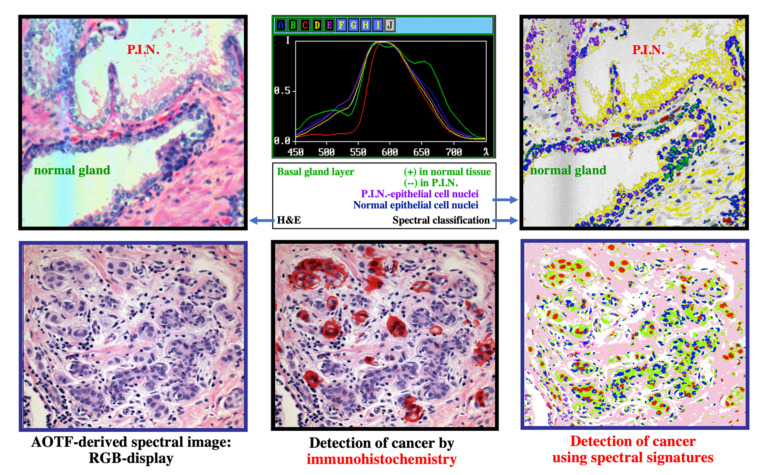
Spectral histopathology. Top: Prostate intraepithelial neoplasia (PIN). Left: RGB image of H&E specimen, difficult to evaluate. Middle: spectral signatures of different specimen areas. Right: spectrally segmented version of H&E image, showing normal gland (with a basal gland layer, in green pseudocolor), and PIN (without the basal layer). Bottom: Breast cancer. Left: RGB image of breast tissue (with a distant tumor that has infiltrated this area, in a difficult to assess way. Middle: Immunohistochemistry staining showing (in red) the cancer areas. Right: The same areas are identified and highlighted as cancerous based on spectral segmentation of just the H&E images (no extra staining needed).

**Figure 10 molecules-26-06651-f010:**
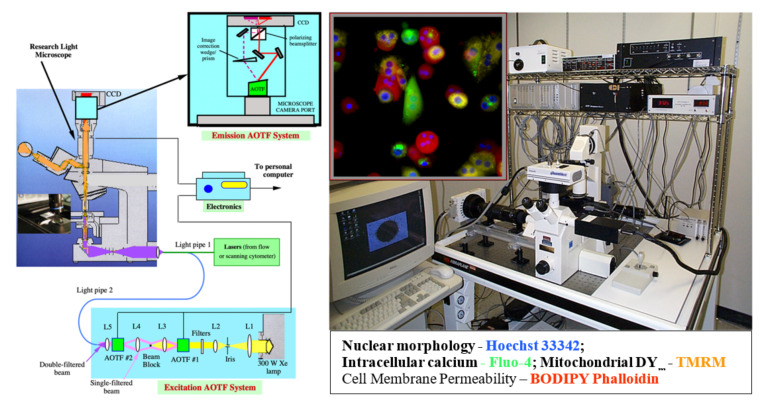
Two-AOTF microscopy workstation: schematics (**left**) shown in a laboratory (**right**). Multimode microscopy-based multiparameter fluorescence cellular studies of drug candidates, in collaboration with Pfizer and JPL (see text for details). The upright microscope in the schematics (**left**) was replaced with an inverted microscope (see image on the right).

**Figure 11 molecules-26-06651-f011:**
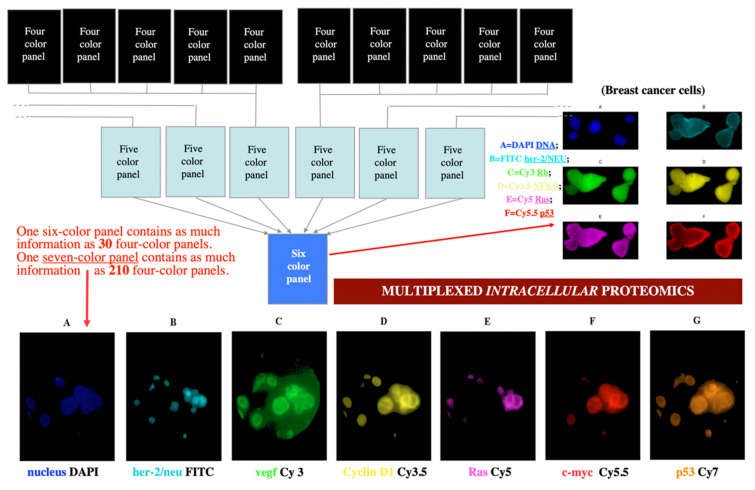
Four to seven-parameter imaging of breast cancer cells. In the most informative case (bottom) images (left to right) are of cells labeled simultaneously with seven different probes: (**A**) the nucleus labeled with DAPI, (**B**) her-2/neu labeled with FITC, (**C**) vegf labeled with Cy3, (**D**) Cyclin D1 labeled with Cy3.5, (**E**) ras labeled with Cy5, (**F**) c-myc labeled with Cy5.5, and (**G**) p53 labeled with Cy7. These were acquired using four separate filter sets (DAPI/FITC/Cy3/Cy5-Cy7) and an AOTF module was used to distinguish between the dyes contained within the filter sets. Up to 10 labels/parameters could be imaged in similar experiments. Spectra from the individually labeled preparations were used as reference spectra to extract the relative amount of each component in the multiply labeled specimen using standard analysis such as least squares.

**Figure 12 molecules-26-06651-f012:**
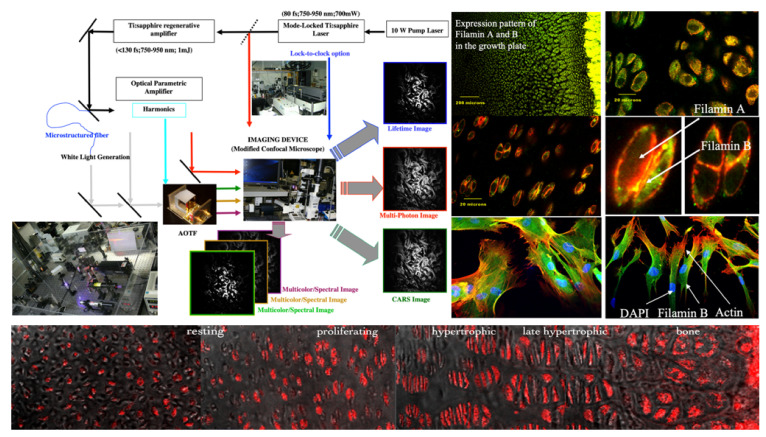
Multispectral microscopy setup and expression pattern of Filamins A and B in the growth plate. Topologically, filamin A (green) is distributed mainly in the hypertrophic zone and in the cytoplasm, while filamin B (red, bottom) is distributed along the growth plate, and in the cleavage furrow of dividing cells.

**Figure 14 molecules-26-06651-f014:**
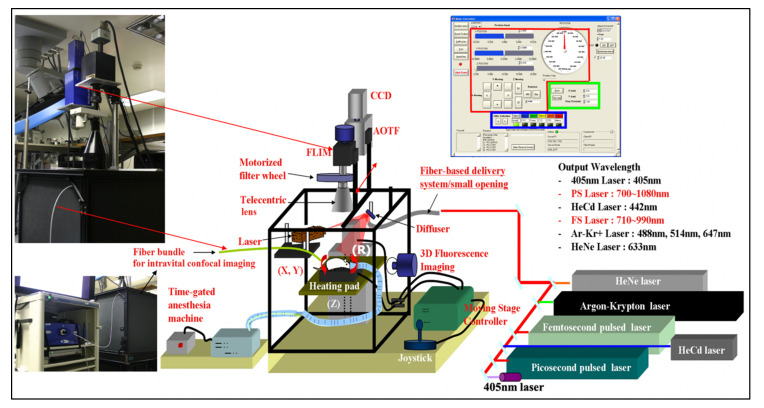
Multimode optical imaging system schematic, with pictures of key components [100,107]. The instrument is capable of several imaging modes, including fluorescence intensity/ratiometric fluorescence, spectral, lifetime, intravital confocal, two-photon, 3D, and bioluminescence imaging. The setup has 7 different lasers, allows for two-photon imaging in three modes (scanning, mosaic, and full-field), and has some other critical components (AOTFs, heating pad, and gated anesthesia for the animals, 3D/rotational stage allowing positioning of the animal under study, telecentric lenses, custom diffusers), mostly housed in a light-tight enclosure.

**Figure 15 molecules-26-06651-f015:**
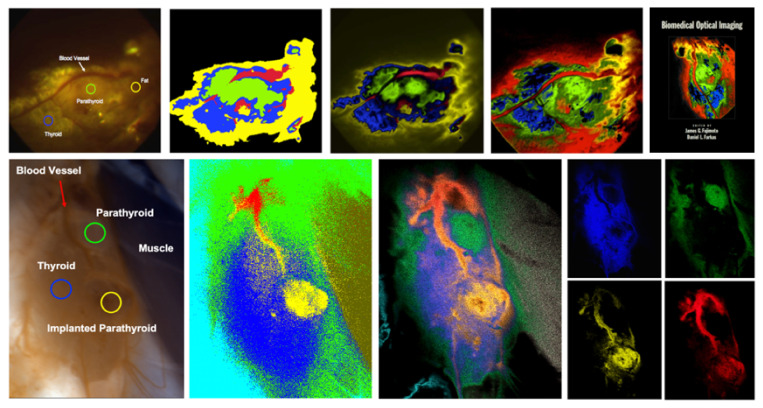
Intelligent spectral signature analysis and display options for in vivo tissue [64,119].

**Figure 16 molecules-26-06651-f016:**
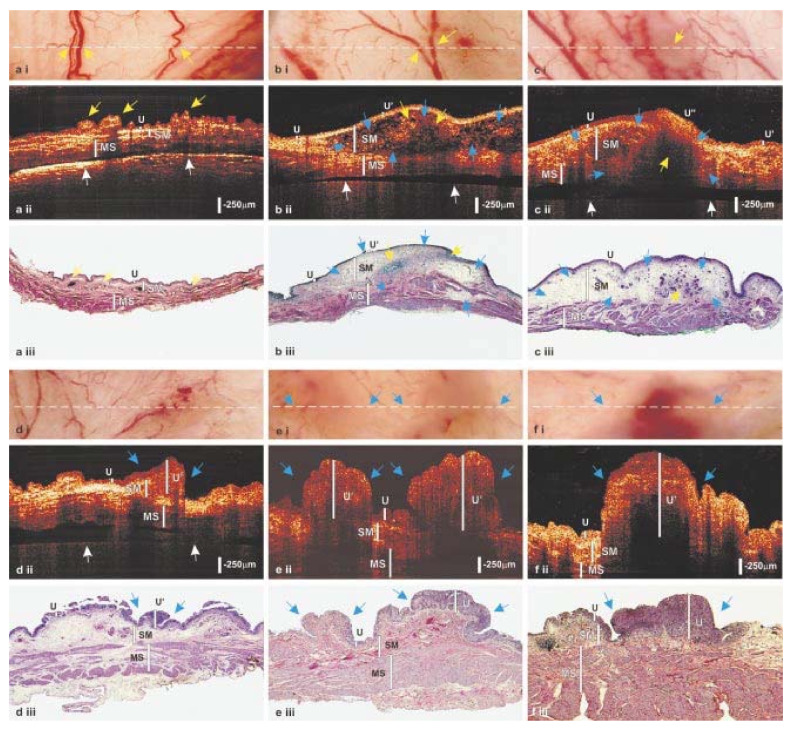
Ex vivo 2D images of rat bladders during tumorigenesis induced by MNU instillation [126]. At each time point, Panels (**i**), (**ii**), (**iii**) (top to bottom) are surface (video) RGB, OCT, and H&E-stained histology, respectively. Image size: X/4.2 mm, Z/1.9 mm. For normal control (group (**a**)), the low scattering (U) and the underlying high scattering SM and MS layers are identified and indicated by white bars. Yellow and white arrows point to superficial blood vessels and the supporting silicon pad, respectively. At 24 h after MNU instillation (group (**b**)), acute chemical cystitis involving inflammatory infiltrate with SM fluid buildup is identified by OCT and evidenced by histology, whereas surface imaging fails to provide any specific feature. The whole-mount bladder wall is thickened. At 48 h (group (**c**)), continued damage to the bladder involving inflammatory infiltrate with SM vasodilation or blood congestion within the SM blood vessels (e.g., severe edema) is identified by OCT and evidenced by histology. Blue arrows point to the SM inflammatory lesion, yellow arrows point to blood congestion. (U”) shows the completely denudated urothelium following MNU installation. At 20 weeks (group (**d**)), in addition to inflammatory infiltrate, chronic damage, e.g., early papillary hyperplasia is identified by OCT as thickened urothelium and evidenced by histology. Blue arrows point to the thickened urothelium (U’). The underlying bladder wall, e.g., SM and MS are still visible. At 30 weeks (group (**e**)), more severe damage is identified by OCT as indicated by blue arrows and evidenced by histology. At 40 weeks (group (**f**)), thickened urothelium and increased vascularization are identified by OCT and evidenced by histology. High magnification histology provides subcellular evidence of the growth of an incipient papillary carcinoma.

**Figure 17 molecules-26-06651-f017:**
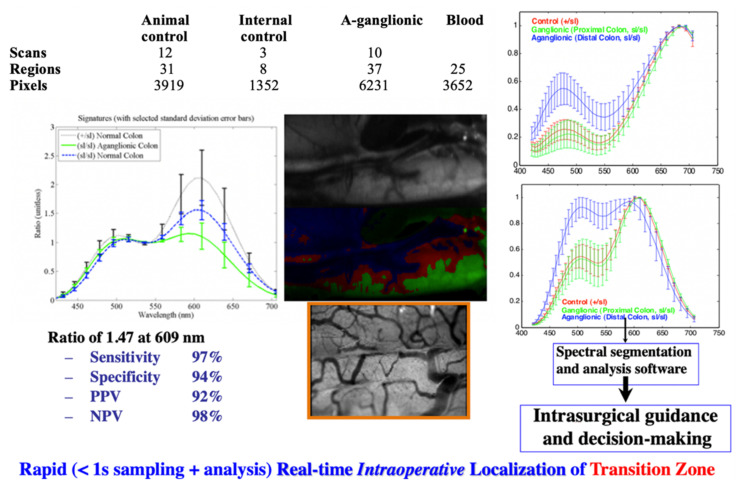
Delineating Hirschsprung’s Disease intraoperatively in an animal model by spectral imaging [128]. Spectral reflectance differences between normal and aganglionic (Hirschsprung-diseased) colon yielded (fast and with very high confidence) a separation between these regions. The (pseudocolor) red transition zone is where the surgical cut is recommended, in near-real-time.

**Figure 19 molecules-26-06651-f019:**
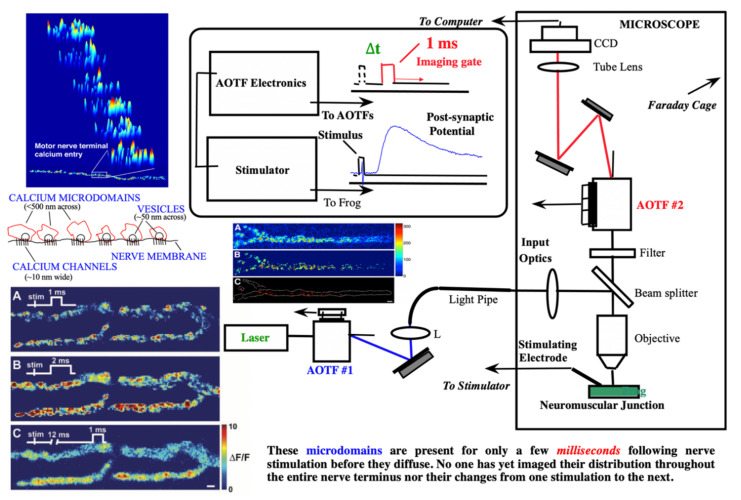
Calcium microdomain dynamics (**left**) imaged with a custom AOTF microscope (**right**) [150]. Left (bottom): The observed spatial profile of Ca^2+^-influx is dependent on the timing of the laser illumination window. (**A**). Representative difference image of Ca^2+^-entry with an illumination window of 1msec duration beginning 1.5 ms after nerve trunk stimulation. (**B**). Representative difference image of Ca^2+^-entry with an illumination window of 2 msec duration beginning 1.5 msec after nerve stimulation. The Ca^2+^-entry signal is more intense (resulting from a doubling in the dye illumination time) but also more diffusely distributed throughout the nerve terminal. (**C**). Representative raw difference image of Ca^2+^-entry with an illumination window of 1 msec duration beginning 12 msec after nerve stimulation. The Ca^2+^-entry signal is slightly reduced in magnitude as compared with *A* and much more diffusely distributed. The timing of the illumination window is shown schematically in each image. The pseudocolor scale bar is the same for all panels and is expressed as ΔF/F(%). Scale bar, 2 µm. Left (top): Upon each consecutive stimulation (diagonally), different domains are activated.

**Figure 20 molecules-26-06651-f020:**
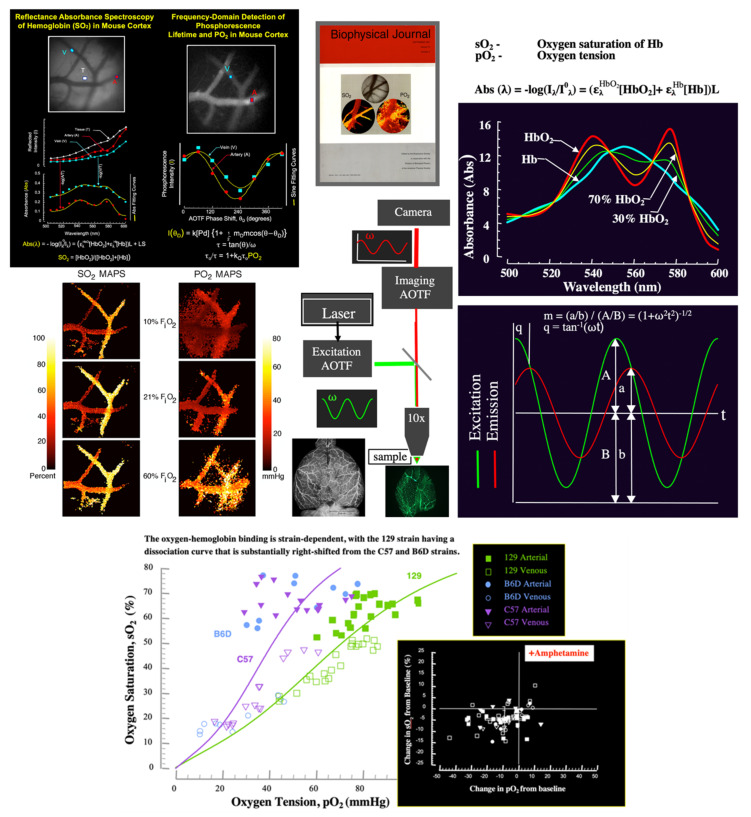
Simultaneous mapping of oxygen saturation of hemoglobin and oxygen tension [150,151]. Method schematics (**top**, **right**); arteries and veins could easily be distinguished and changed upon inhalation O_2_ concentration being changed (**top**, **left**). Sigmoidal pO_2_ vs. sO_2_ curves were obtained, different between mouse lines (**left**). Amphetamine decreased both sO_2_ and pO_2_.

**Figure 21 molecules-26-06651-f021:**
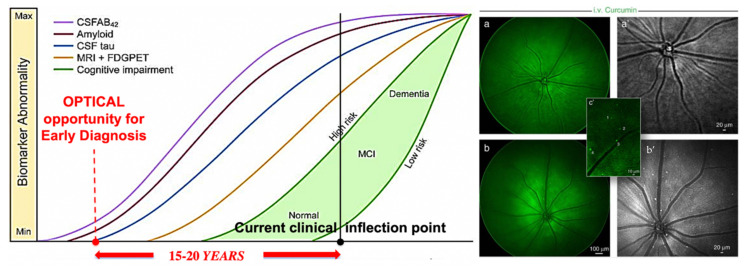
Retinal Optical Imaging: using biomarkers for very early detection of Alzheimer’s Disease. Left (modified from [155]): If a proper AD biomarker is identified and is assessable (via high-resolution imaging) when it is significantly smaller than the current detection limit by any other method, insights could be gained on early pathology that precedes disease symptoms by as much as 20 years, thus opening new interventional possibilities; Right: representative imaging data in living mice retinas, showing no plaques ((**a**,**a’**), top) and early plaques ((**b**,**b’**), bottom) at higher resolution ((**c’**), insert), all highlighted by fluorescence of (intravenously injected) curcumin [157].

**Figure 22 molecules-26-06651-f022:**
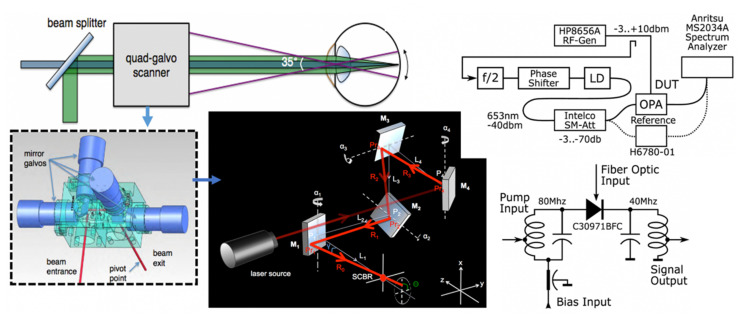
New multimode confocal scanning laser ophthalmoscope. The system is based on a supercontinuum laser light source with spectral options (not shown), quad-galvo scanning (**left**) [161], and degenerate optical parametric amplification-based low light detection (**right**) [162].

**Figure 23 molecules-26-06651-f023:**
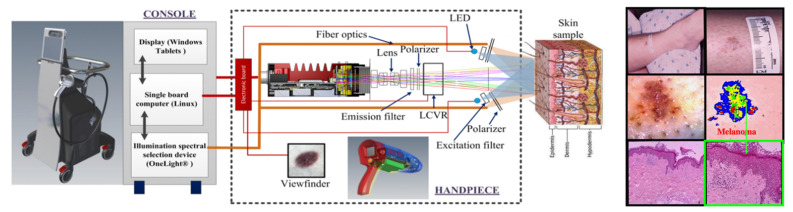
Multimode imaging dermoscope (SkinSpect) (**Left**) and some early melanoma detection results (**Right**).

**Figure 24 molecules-26-06651-f024:**
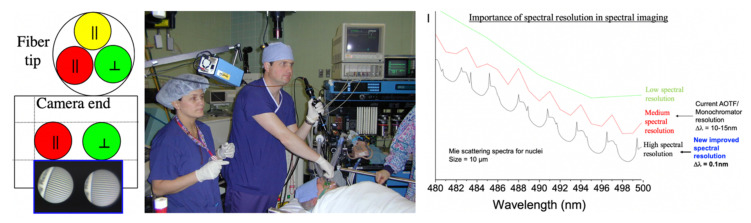
Elastic scattering imaging endoscopy. (**Left**): custom fiber distributions and polarizations; (**Middle**): The workstation during an endoscopic procedure at AGH, Pittsburgh; (**Right**): Improvement in spectral resolution shows true Mie scattering oscillation patterns.

**Figure 25 molecules-26-06651-f025:**
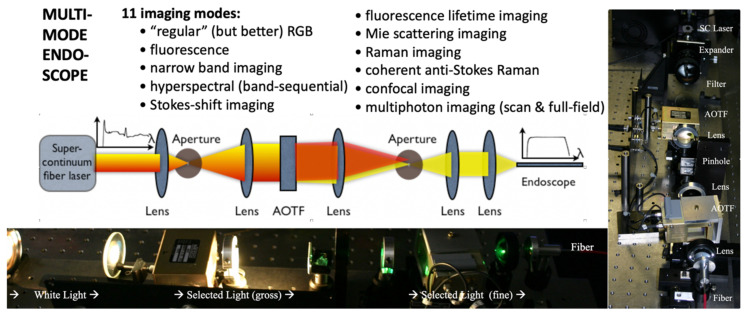
Multimode imaging system for minimally invasive surgery (see text for details).

**Figure 26 molecules-26-06651-f026:**
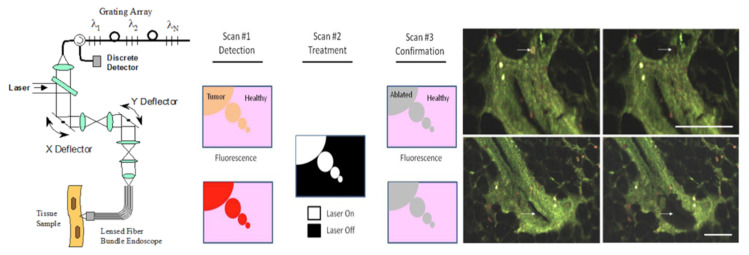
Theranostic high-speed multispectral confocal biomedical imaging and treatment [207,208,209,210,211]. (**Left**): schematics of multispectral scanning confocal system (endoscopy mode). (**Middle**): theranostic intervention by spectral confocal imaging (Scan #1) segmenting the imaged area into healthy and tumor (red); Scan #2 tumor is ablated with pixel precision, by firing the higher-powered laser only over pixels segmented as tumor (with any safety margin chosen). Scan #3 confirms the ablated region. (**Right**): examples of treatment in rat breast cancer model (scale bars = 100 µm)—arrows indicate where cancer was found (left panels) and ablated (right panels).

**Figure 27 molecules-26-06651-f027:**
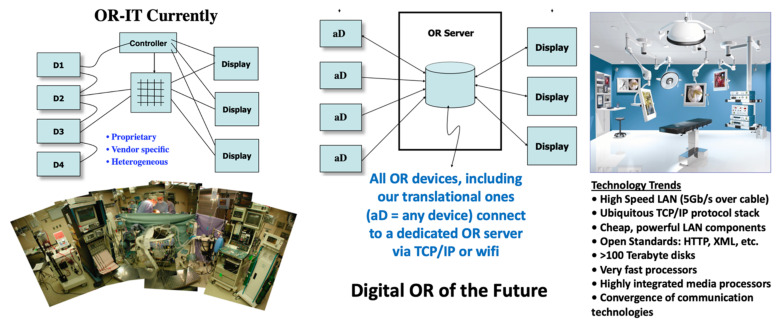
Current (**left**) OR vs. our proposed Digital OR of the Future (**right**) (Nowatzyk, A.G and Farkas, D.L.).

**Figure 28 molecules-26-06651-f028:**
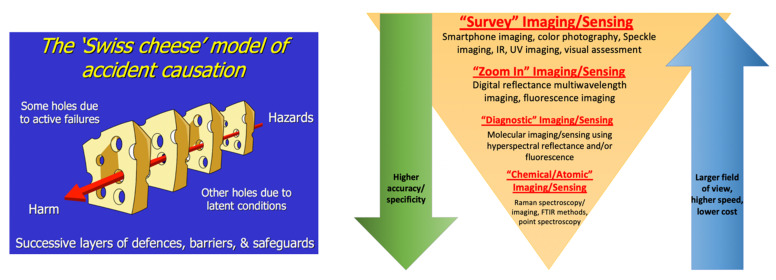
Risk (disease/problem non-detection) mitigation by “stacked, funneled” multimode imaging.

**Table 1 molecules-26-06651-t001:** Multimode assessment of nanoconstructs and HerGa chemotherapy (top) and application and capabilities of each imaging mode in the multimode panel (bottom) [100,101,102,103,104,105,106,107,108,109,110,111,112,113,114,115].

Application	Combined Imaging Modes	Information Obtained
•Nanoconstruct therapy	Fluorescence intensity Spectral imaging	1. Dynamic monitoring of nanoconstruct distribution and examination of tumor targeting capability 2. Examination of nanoconstruct clearance 3. Nanoconstruct effects on specific organs
•HerGa chemotherapy	Fluorescence intensity Spectral and lifetime imaging Scanning/full field two photon- excited fluorescence imaging	1. Dynamic monitoring of HerGa and S2Ga distributions and tumor targeting capability assessment 2. Examination of HerGa accumulation kinetics 3. Mechanism of HerGa action and effects on organs 4. Tumor environment information 5. Usefulness of HerGa in surgical tumor detection
**Imaging Mode**	**Biomedical Application**	**Capability Derived**
•Fluorescence intensity	Measurement of relative accumulated concentration and discrimination of 2 fluorophores (AF680 and Rh123) conjugated to drug molecules in nude mice	Kinetics/Dynamics
•Spectral imaging	Quantitative discrimination of fluorophores (fluorescein and corroles) from autofluorescence in nude mice	Quantification
•Fluorescence lifetime imaging	Monitoring functional status in vicinity of fluorophores and quantitative discrimination between them	Environment assessment Quantification
•Intravital confocal imaging	Observation of microstructures such as muscle fibers and blood vessels without biopsy or staining	High resolution/ magnification
•Scanning/widefield 2-photon excited fluorescence imaging	High magnification/resolution observation of intact tissues (tumor regions) inside small animals in vivo and ex vivo (tumors, livers, eyeballs of AD mice)	High resolution/ magnification
•Bioluminescence imaging	Detection of ATP and enzymatic activity in engineered nude mice	High sensitivity

**Table 2 molecules-26-06651-t002:** Advanced imaging for specificity & sensitivity in our pre-clinical * and clinical ** work.

	Sensitivity (%)	Specificity (%)	Mode	References
**1. Cytopathology **** (Papanicolau test)	**97**	**99**	**S**	[65,66,67,68]
**2. Histopathology **** (H&E, immunohistochem.)	**98**	**96**	**S**	[59,60,61,62,63,64,165,166,167]
**3. Tissue oxygenation mapping ***	**100**	**100**	**M2**	[73,150,151]
**4. Breast cancer *** (no contrast agent) *Note: Second set with more advanced analysis.* *Note: With multimode acquisition/analysis*	**91** **96** **96**	**86** **92** **97**	**S** **S** **M7**	[117] [117] [116,118]
**5. Lymph nodes in vivo *** (BC metastasis assess.)	**100**	**100**	**M4**	[120,121]
**6. Hirschsprung’s Disease in vivo *** *Note: Second set with more advanced analysis*	**97** **98**	**94** **98**	**S** **S**	[127,128,129] [128]
**7. Alzheimer’s Disease *^,^****	**100**	**100**	**M3**	[156,157,158,159]

In the “Mode” column, S denotes spectral/hyperspectral, while M signifies multimode, with the number following denoting the number of modes that came together on the instrument for that particular study/application. Note: some of these % results are based on a small number of cases, and thus should be treated with caution.

## Data Availability

No new data were created or analyzed in this review study. Data sharing is not applicable to this article.

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
