# Peer review of "Biomedical Applications of Translational Optical Imaging: From Molecules to Humans"

_molecules, 2021, doi:10.3390/molecules26216651_

Round 1
Reviewer 1 Report
Optical imaging techniques are widely used in biomedicine. Their great diversity and rapid development can be barely tracked and analyzed. That is why the reviews on the modern achievements in translational optical imaging are worth writing. In my opinion, any review paper should provide an overview of the current state of knowledge and contain a substantial analysis of existing methods, their classification and comparison. In a current state, this manuscript looks like a collection of author’s results. It includes more than 150 self-citations and ignores the results of other researchers. Therefore, I cannot recommend publication of this manuscript in Molecules.
Author Response
Please see the attachment entitled "Responses to reviewers of molecules - 1206712". Thank you.

Reviewer 2 Report
Farkas reviews a considerable corpus of literature giving a personal yet authoritative point of view on the field of optical imaging.
I would gladly suggest this work for acceptance in Molecules.
As minor comments:
-introducing the review with a sentence such as "focusing heavily on our own work, merely because we understand it best" is quite a strong statement and is extremely subjective. Consider removing it.
-check references at line 845 and 1519
-consider remaking some of the figures (e.g. Fig 3): the letters in the figures are extremely tiny and basically unreadable
-
Author Response
Please see the attachment entitled "Response to reviewers of moleculaes-1206712" Thank you.

Round 2
Reviewer 1 Report
Seems like there really must have been some miscommunication. If you were invited to describe your own long-term results, then it is ok. I just would like to ask you to state clearly in the abstract that this a review of your results mainly.
I have doubts about the quantitative data on the sensitivity and specificity of spectral imaging in medical practice (Table 2). I cannot agree with you when you declare up to 100% based on only on your own research. There is a lot of review and research papers dedicated to this topic. I think table 2 should be revised and well commented, and include more references.
Author Response
Thank you for re-reviewing this manuscript.
I have made the changes you requested to the abstract, as well as to both the title and endnote of Table 2. I have also tightened the language and some details in the body of the manuscript and added a number of really recent references at the end of the Discussion. I hope that all these, taken together, will prove satisfactory.